# Competition between myosin II and β_H-spectrin regulates cytoskeletal tension

Consuelo Ibar[1], Krishna Chinthalapudi[2], Sarah M Heissler[2], Kenneth D Irvine[1]*

[1]Waksman Institute and Department of Molecular Biology and Biochemistry, Rutgers University, Piscataway, United States; [2]Department of Physiology and Cell Biology, Dorothy M. Davis Heart and Lung Research Institute, The Ohio State University College of Medicine, Columbus, United States

*For correspondence:
irvine@waksman.rutgers.edu

Competing interest: The authors declare that no competing interests exist.

**Abstract** Spectrins are membrane cytoskeletal proteins generally thought to function as hetero-tetramers comprising two α-spectrins and two β-spectrins. They influence cell shape and Hippo signaling, but the mechanism by which they influence Hippo signaling has remained unclear. We have investigated the role and regulation of the *Drosophila* β-heavy spectrin (β_H-spectrin, encoded by the *karst* gene) in wing imaginal discs. Our results establish that β_H-spectrin regulates Hippo signaling through the Jub biomechanical pathway due to its influence on cytoskeletal tension. While we find that α-spectrin also regulates Hippo signaling through Jub, unexpectedly, we find that β_H-spectrin localizes and functions independently of α-spectrin. Instead, β_H-spectrin co-localizes with and reciprocally regulates and is regulated by myosin. *In vivo* and *in vitro* experiments support a model in which β_H-spectrin and myosin directly compete for binding to apical F-actin. This competition can explain the influence of β_H-spectrin on cytoskeletal tension and myosin accumulation. It also provides new insight into how β_H-spectrin participates in ratcheting mechanisms associated with cell shape change.

## eLife assessment

The manuscript provides **valuable** insights into the regulatory role and mechanisms of the spectrin cytoskeleton in mechanotransduction in *Drosophila*. The data are **compelling** in establishing that alpha and beta spectrin regulate the Hippo signaling pathway independently via their effect on cytoskeletal tension. The work will be of interest to cell and developmental biologists, particularly those who focus on mechanotransduction and the cytoskeleton.

## Introduction

The spectrin cytoskeleton has been described as a lattice of cross-linked, spring-like proteins that provide structural support to cells (*Liem, 2016*; *Machnicka et al., 2014*). Spectrins were first discovered and characterized in red blood cells but are expressed in many cell types. Spectrins can bind to cell membranes and F-actin, linking them together. They are generally thought to act as hetero-tetramers, composed of two α subunits and two β subunits. *Drosophila* has one α-spectrin (α-Spec) and two β-spectrins: β-spectrin (β-Spec) and β-heavy spectrin (β_H-Spec, encoded by *karst* [*kst*]), which could potentially generate two distinct spectrin heterotetramers: (αβ)_2 and (αβ_H)_2. β-Spec and β_H-Spec interact with F-actin through their N-terminal domains, which contain two actin-binding calponin-homology (CH) domains, connecting the spectrin cytoskeleton to the actin cytoskeleton (*Liem, 2016*). In *Drosophila* epithelia, it has been reported that β-Spec localizes to the lateral sides of cells, β_H-Spec localizes to the apical sides of cells, and α-Spec localizes both laterally and apically, leading to inferences that spectrin exists as lateral (αβ)_2 tetramers and apical (αβ_H)_2 tetramers (*Dubreuil et al.,*

*1997*; *Lee et al., 1997*; *Thomas et al., 1998*; *Zarnescu and Thomas, 1999*). Despite common assumptions that spectrins act as tetramers, there is some evidence that alternative arrangements may exist. In *Drosophila* ovarian follicle cells, the absence of α-Spec diminishes the recruitment of β$_H$-Spec to the apical domain, but it does not affect the recruitment of β-Spec to the lateral domain (*Lee et al., 1997*). Examination in *Drosophila* of a mutant form of α-Spec that, based on *in vitro* studies, is unable to bind β-Spec and compromised in its ability to bind β$_H$-Spec revealed that it could nonetheless rescue the lethality of an *α-Spec* mutant (*Khanna et al., 2015*). Experiments done with a mammalian homolog of β$_H$-Spec, βV-Spec, showed that it can homodimerize through its C-terminal region, raising the possibility that βV-Spec might be able to cross-link F-actin by itself (*Papal et al., 2013*).

Several studies have reported that spectrins also regulate Hippo signaling, with effects on readouts of Hippo signaling reported in *Drosophila* imaginal discs and ovarian follicles, as well as in cultured mammalian cells (*Deng et al., 2015*; *Deng et al., 2020*; *Fletcher et al., 2015*; *Wong et al., 2015*). Hippo signaling is a signal transduction network that responds to diverse upstream inputs, including the cytoskeleton and cells' physical environment (*Misra and Irvine, 2018*; *Zheng and Pan, 2019*). Hippo signaling modulates cell proliferation and fate, largely through the regulation of Yap family transcriptional co-activator proteins (Yorkie, Yki, in *Drosophila*, YAP1 and TAZ in humans). Yki is primarily regulated through phosphorylation by the kinase Warts (Wts), which promotes the cytoplasmic localization of Yki. Various potential mechanisms for biomechanical regulation of Yki/Yap activity have been described, but the best-characterized mechanism in *Drosophila* is the Jub biomechanical pathway. This involves tension-dependent recruitment of an Ajuba LIM protein (Jub in *Drosophila*, LIMD1 in mammals) to α-catenin at adherens junctions (AJs) (*Alégot et al., 2019*; *Ibar et al., 2018*; *Rauskolb et al., 2022*; *Rauskolb et al., 2014*; *Sarpal et al., 2019*; *Sun et al., 2015*). Jub then recruits and inhibits Wts, resulting in increased Yki activity.

Studies of the influence of spectrins on Hippo signaling have suggested different mechanisms by which this might occur (*Deng et al., 2015*; *Deng et al., 2020*; *Fletcher et al., 2015*; *Wong et al., 2015*). *Fletcher et al., 2015*, focusing on β$_H$-Spec, suggested that the spectrin cytoskeleton influences the membrane density, and thereby the activation state, of upstream regulators of Hippo signaling. *Wong et al., 2015*, focusing on β-Spec, suggested that spectrins might influence Hippo signaling by modulating levels of F-actin; increased levels of F-actin have been reported in other studies to be associated with increased Yki/Yap activity (*Aragona et al., 2013*; *Fernández et al., 2011*; *Sansores-Garcia et al., 2011*). *Deng et al., 2015*, focusing on α-Spec, reported that spectrins regulate levels of phosphorylated myosin light chain (required for myosin activation) but surprisingly did not affect levels of myosin or recruitment of Jub, leading them to infer that spectrins can act through a novel tension-dependent but Jub-independent pathway. These same authors later reported that in the pupal eye, spectrins can act through Jub, while suggesting that the action of spectrins in the pupal eye differs from their action in the wing disc (*Deng et al., 2015*; *Deng et al., 2020*).

The disparate models for how spectrins influence Hippo signaling have led to confusion over whether a distinct spectrin-based mechanism exists for the mechanical regulation of Hippo signaling. We were particularly interested in investigating claims that spectrins could alter cytoskeletal tension in wing discs without affecting Jub localization or levels of myosin. Our results reveal that both β$_H$-Spec and α-Spec influence Jub recruitment to AJ, and their effects on Yki activity depend upon Jub. Together, these observations argue that spectrins influence Hippo signaling through the Jub biomechanical pathway. Unexpectedly, our investigations also reveal that β$_H$-Spec and α-Spec act separately in wing discs - they do not co-localize, nor do they influence each other's localization. These observations argue that β$_H$-Spec does not act as part of an (αβ$_H$)$_2$ tetramer in the wing disc but rather exerts its functions independently of α-Spec. Finally, we establish that β$_H$-Spec and myosin reciprocally antagonize each other's apical localization in wing discs - myosin inhibits β$_H$-Spec, and β$_H$-Spec inhibits myosin. We further show that myosin can compete with β$_H$-Spec for binding to F-actin *in vitro*. Together with structural modeling, our observations argue that β$_H$-Spec and myosin compete with each other for binding to F-actin *in vivo*. This competition could explain how β$_H$-Spec influences myosin activity and further suggests a simple mechanism for the contribution of β$_H$-Spec to ratcheting processes that alter cell shape via actomyosin contractility.

# Results

## β$_H$-Spec regulates myosin activity and levels in wing imaginal discs

Prior studies have reported that mutation or RNAi-mediated knockdown of spectrins in wing and eye imaginal discs increased myosin activity, as visualized by staining for myosin light chain phosphorylated at activation sites (pMLC) (*Deng et al., 2015*; *Forest et al., 2018*). Surprisingly, it was also reported that levels of GFP-tagged myosin light chain (encoded in *Drosophila* by spaghetti squash, Sqh:GFP) or F-actin were nonetheless unaffected (*Deng et al., 2015*), whereas changes in myosin activity generally correlate with changes in myosin accumulation (*Fernandez-Gonzalez et al., 2009*; *Noll et al., 2017*). To re-examine this, we analyzed wing imaginal discs in which the apical, β$_H$-Spec *kst* was knocked down in posterior wing cells by expressing UAS-RNAi lines under *en-Gal4* control (RNAi validation is presented in *Figure 1—figure supplement 1*). This approach leaves unaffected anterior wing disc cells as an internal control. These experiments typically also include a neutral transgene expressed under en-Gal4 control to mark posterior cells (e.g. UAS-RFP) and a transgene expressing Dicer2 (Dcr2) to increase the efficacy of RNAi (*Dietzl et al., 2007*). This increased apical pMLC in posterior cells, where levels of β$_H$-Spec were reduced (*Figure 1—figure supplement 2B and D*), consistent with previous reports (*Deng et al., 2015*; *Forest et al., 2018*). A difference in junctional tension between anterior (control) cells and posterior (β$_H$-Spec RNAi) cells in our experiments was confirmed by measuring the recoil velocity of cell junctions after laser cutting, which demonstrated an increased tension in the β$_H$-Spec depleted sides (*Figure 1G*). To examine myosin protein levels, we employed the myosin light chain GFP fusion Sqh:GFP. We found that depletion of β$_H$-Spec led to increased levels of junctional myosin (*Figure 1B*). To quantify these effects on Sqh:GFP levels, we made maps of Sqh:GFP intensity normalized against E-cad intensity. These are displayed on a red (high) to blue (low) heat map (*Pan et al., 2018*) and we also used this analysis to calculate the ratio of intensities of the anterior (control) versus posterior (experimental) compartments (*Alégot et al., 2019*; *Figure 1D–E, H–I*). This was further confirmed by using a distinct β$_H$-Spec RNAi line (*UAS-kstRNAi[HMS00882]*), which similarly increased myosin levels (*Figure 1—figure supplement 2E*).

As earlier studies reporting no effect on myosin levels focused on mutation or knockdown of α-Spec (*Deng et al., 2015*), we considered the possibility that knockdown of α- and β$_H$-Spec might differ in their effects on apical myosin accumulation. However, we found that knockdown of α-Spec also caused an increased accumulation of apical myosin, as well as pMLC and junctional tension, similar to the effects of β$_H$-Spec knockdown (*Figure 1C, F, H, I*; *Figure 1—figure supplement 2C and D*). α-Spec and β$_H$-Spec knockdown differed though in that the thickness of the wing disc epithelium was reduced by α-Spec knockdown, but not by β$_H$-Spec knockdown (*Figure 1—figure supplement 3*), indicating that even though α-Spec and β$_H$-Spec have similar effects on junctional myosin, their roles in wing disc cells differ. Studies in pupal eyes identified a role for α-Spec in attaching F-actin to membranes that was associated with maintaining proper cortical tension and cell shape (*Deng et al., 2020*); we think the reduced thickness of the epithelium is a reflection of this role in wing discs. That is, reduced stiffness of the lateral sides of cells when α-Spec is removed may allow cells to expand laterally, and through conservation of volume, simultaneously shorten along the apical-basal axis.

## β$_H$-Spec regulates Hippo signaling through Jub

It was previously argued that spectrins do not influence Hippo signaling through the Jub biomechanical pathway in wing discs, in part based on an apparent lack of effect of α-*Spec* mutation or depletion on Jub localization (*Deng et al., 2015*). However, multiple studies have consistently observed that Jub localization increases when tension at AJ is increased (*Forest et al., 2018*; *Pan et al., 2018*; *Rauskolb et al., 2019*; *Rauskolb et al., 2014*; *Razzell et al., 2018*). Thus, we examined Jub localization under conditions of β$_H$-Spec depletion. In wing imaginal discs, Jub accumulates in puncta that often occur near intercellular vertices, together with a lower level, more even accumulation along the cell-cell junctions (*Figure 2A* and *Figure 2—figure supplement 1B*). Jub is recruited by a tension-dependent conformational change of α-catenin, and Jub puncta are increased when cytoskeletal tension is higher. Consistent with this, examination of Jub:GFP confirmed that the increased junctional tension and myosin activity caused by β$_H$-Spec depletion is associated with increased junctional recruitment of Jub (*Figure 2B, E and G–H* and *Figure 2—figure supplement 1C*).

To assess the functional significance of increased Jub localization to AJ under β$_H$-Spec knockdown conditions, we assayed the ability of RNAi-mediated *jub* knockdown to suppress β$_H$-Spec (kst) RNAi

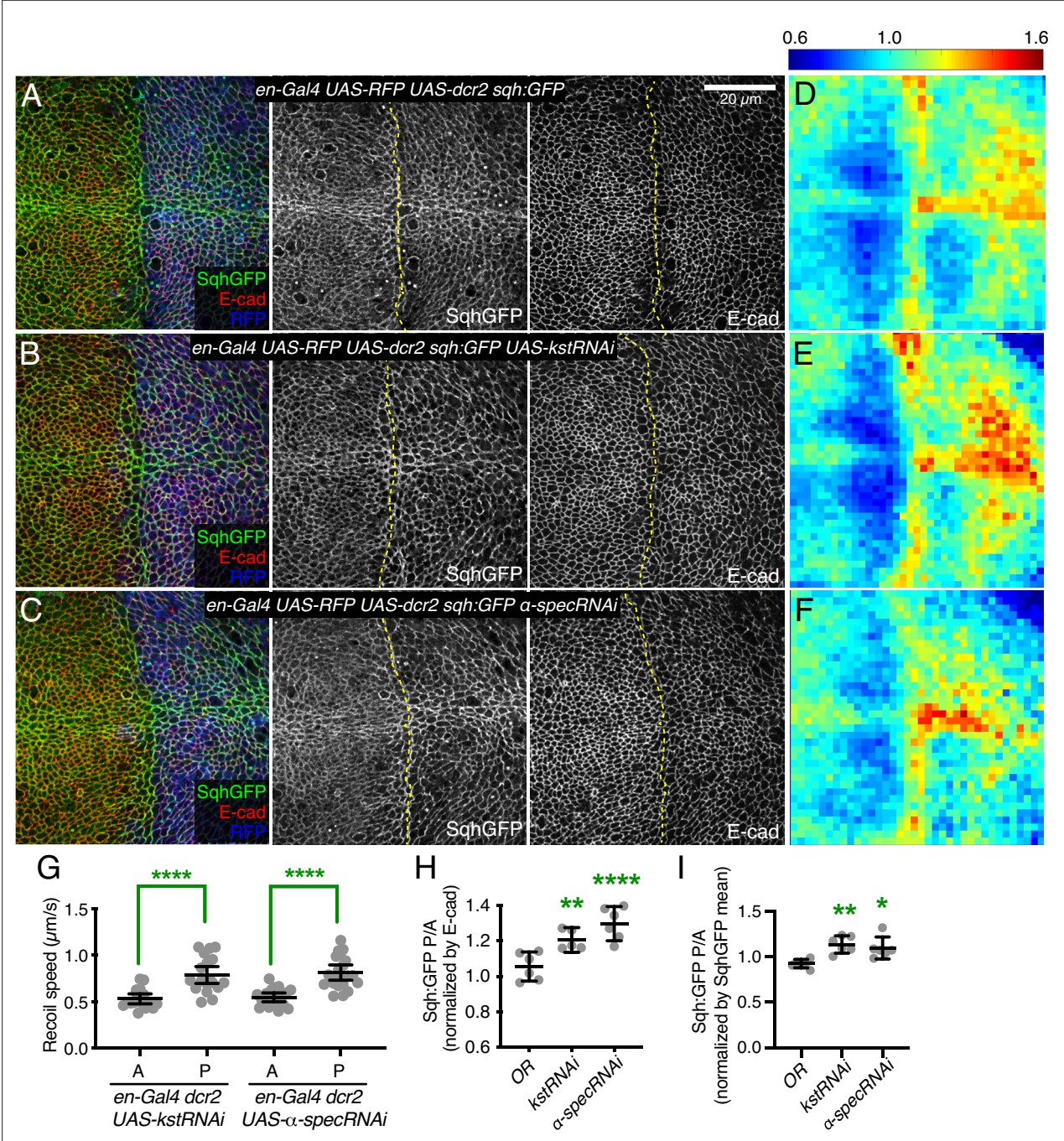

**Figure 1.** Knockdown of β-heavy spectrin (β$_H$-Spec) or α-spectrin (α-Spec) increases junctional myosin levels in wing discs. (**A–C**) Apical sections of wing discs stained for E-cad (red/gray) expressing en-Gal4 UAS-RFP UAS-dcr2 sqh:GFP crossed to control (Oregon-R, OR) (**A**), UAS-kstRNAi(v37075) (**B**), or UAS-α-specRNAi (**C**) showing the effect on Sqh:GFP (green/gray) levels and localization in the posterior compartment (marked by RFP, blue). Dashed yellow line marks A/P compartment boundary. Scale bar = 20 μm; all images are at the same magnification. Panels to the right, in gray, show single channels, as indicated. (**D–F**) Heat maps of relative junctional Sqh:GFP intensity of wing discs. Average levels of Sqh:GFP relative to E-cad levels are shown for the different genotypes analyzed in A–C. Heat map scale is indicated on the top. Number of wing discs used for analysis: Control (OR), n=6; UAS-kstRNAi, n=5; UAS-α-specRNAi, n=6. (**G**) Average recoil velocities after laser cutting of cell junctions in anterior (**A**) or posterior (**P**) compartments of wing discs expressing UAS-kstRNAi or UAS-α-specRNAi in posterior cells under en-Gal4 control (n=20). (**H–I**) Quantification of Sqh:GFP normalized to E-cadherin in posterior cells (**P**) compared to anterior cells (**A**) in wing disc expressing the indicated constructs, displayed as individual values, normalized by E-cad (**H**) or normalized by the mean intensity of Sqh:GFP (**I**). Data are shown as mean±95% CI. Statistical significance in (**G**) was determined by Student's t-test between A and P. For (**H**) and (**I**), statistical significance was determined by a one-way analysis of variance (ANOVA) with Dunnett's multiple comparison test relative to the control (Oregon-R): ns: not significant, *p<0.05; **p≤0.01; ***p≤0.001; ****p≤0.0001.

*Figure 1 continued on next page*

*Figure 1 continued*

The online version of this article includes the following figure supplement(s) for figure 1:

**Figure supplement 1.** Validation of RNAi lines.

**Figure supplement 2.** Modulation of levels of total and active myosin in wing discs by spectrins.

**Figure supplement 3.** α-Spectrin (α-Spec) knockdown decreases wing disc thickness.

phenotypes. Knockdown of β$_H$-Spec throughout the developing wing under *nub-Gal4* control increases wing size (wings were 116% and 119% of *nub-Gal4 UAS-dcr2* control size for the two different UAS-RNAi lines used) (*Figure 3A, C–D, and K*; *Deng et al., 2015*; *Fletcher et al., 2015*). Knockdown of *jub* leads to smaller wings (*Das Thakur et al., 2010*; *Rauskolb et al., 2011*). In animals with simultaneous RNAi knockdown of *jub* and *kst,* wing size is similar to that of *jub* RNAi wings (*Figure 3B, E–F, and K*). The epistasis of *jub* to *kst* is consistent with the hypothesis that β$_H$-Spec regulates wing size through its tension-dependent regulation of Jub. To further illustrate this, we reduced tension in wing discs with β$_H$-Spec knockdown by simultaneous RNAi knockdown of *Rho kinase (Rok)* (*Rauskolb et al., 2014*; *Winter et al., 2001*). This reduced wing size and junctional Jub localization similar to that observed in control wing discs with *Rok* knockdown (*Figure 3—figure supplement 1*; *Rauskolb et al., 2014*).

To confirm that the relationship between Jub and β$_H$-Spec revealed by analysis of adult wing size corresponds to changes in Hippo pathway activity, we analyzed the expression of a transcriptional reporter of *expanded (ex)*, *ex-lacZ*, which is a direct target of Yki (*Hamaratoglu et al., 2006*). Knockdown of β$_H$-Spec in posterior compartments through the expression of UAS-*kst* RNAi under *en-Gal4* control caused a mild increase in *ex-lacZ* expression compared to the anterior compartment and to control posterior compartments (*Figure 3L and M*; *Fletcher et al., 2015*). Knockdown of *jub* reduces *ex-lacZ* expression (*Das Thakur et al., 2010*; *Rauskolb et al., 2011*). Simultaneous RNAi knockdown of *jub* and *kst* reduced *ex-lacZ* expression, similar to that in *jub* RNAi cells (*Figure 3N and O*). The suppression of the influence of *kst* on Hippo signaling is again consistent with the inference that β$_H$-Spec regulates Hippo signaling through the Jub biomechanical pathway.

As claims that spectrins act independently of Jub in wing discs were based primarily on analysis of α-Spec, we also examined the effect of α-Spec knockdown on Jub levels. When α-Spec was knocked down in posterior cells by *en-Gal4*-driven RNAi, we observed that recruitment of Jub to cell junctions was increased (*Figure 2C and F–H*). Moreover, as for β$_H$-Spec, we found that the increased wing size and *ex-lacZ* expression caused by knockdown of α-Spec were suppressed by knockdown of *jub* (*Figure 3G, H, K, P and Q*). Thus, as for β$_H$-Spec, and as suggested for pupal eyes (*Deng et al., 2020*), our observations imply that α-Spec also regulates Hippo signaling through Jub in wing discs.

## β$_H$-Spec localizes independently from α-Spec in wing disc cells

Prior studies of spectrin function in imaginal discs have assumed that they function as heterotetramers, with an apical complex composed of $(\alpha\beta_H)_2$ subunits and a lateral complex composed of $(\alpha\beta)_2$ subunits (*Deng et al., 2015*; *Deng et al., 2020*; *Fletcher et al., 2015*; *Thomas et al., 1998*), as was originally suggested for ovarian follicle cells (*Lee et al., 1997*). However, our examination of spectrin localization in wing imaginal discs suggested that the apical-most distributions of α-Spec and β$_H$-Spec differ. To directly compare them, we used an antibody against α-Spec (*Dubreuil et al., 1987*) and a fully functional genomic YFP-trap of β$_H$-Spec (Kst:YFP) (*Lye et al., 2014*). β$_H$-Spec is localized, as reported previously, at the apicobasal level of the AJ in wing discs (*Fletcher et al., 2015*; *Forest et al., 2018*; *Figure 4A and C*). Conversely, α-Spec is enriched in a sub-apical region just below this, with slightly lower levels extending all along the lateral membrane (*Figure 4B and C*). Only very low levels of α-Spec are detected in the apical plane where β$_H$-Spec is detected, and their distributions in this plane appear to differ (*Figure 4A*).

To confirm that β$_H$-Spec and α-Spec localize independently of each other in wing discs, we examined the consequences of depleting α-Spec, β-Spec, or β$_H$-Spec on each other's localization. For this, we used *en-Gal4*-driven UAS-RNAi lines to knock down one of the spectrin proteins, and then examined whether the localization of the others was affected. These experiments revealed, as expected, that β-Spec is not required for β$_H$-Spec localization (*Figure 5B*), but it does strongly reduce α-Spec localization (*Figure 5E*). α-Spec is not required for β$_H$-Spec localization (*Figure 5A*) but its knockdown slightly reduces β-Spec localization (*Figure 5D*). Finally, β$_H$-Spec depletion does not affect α-Spec

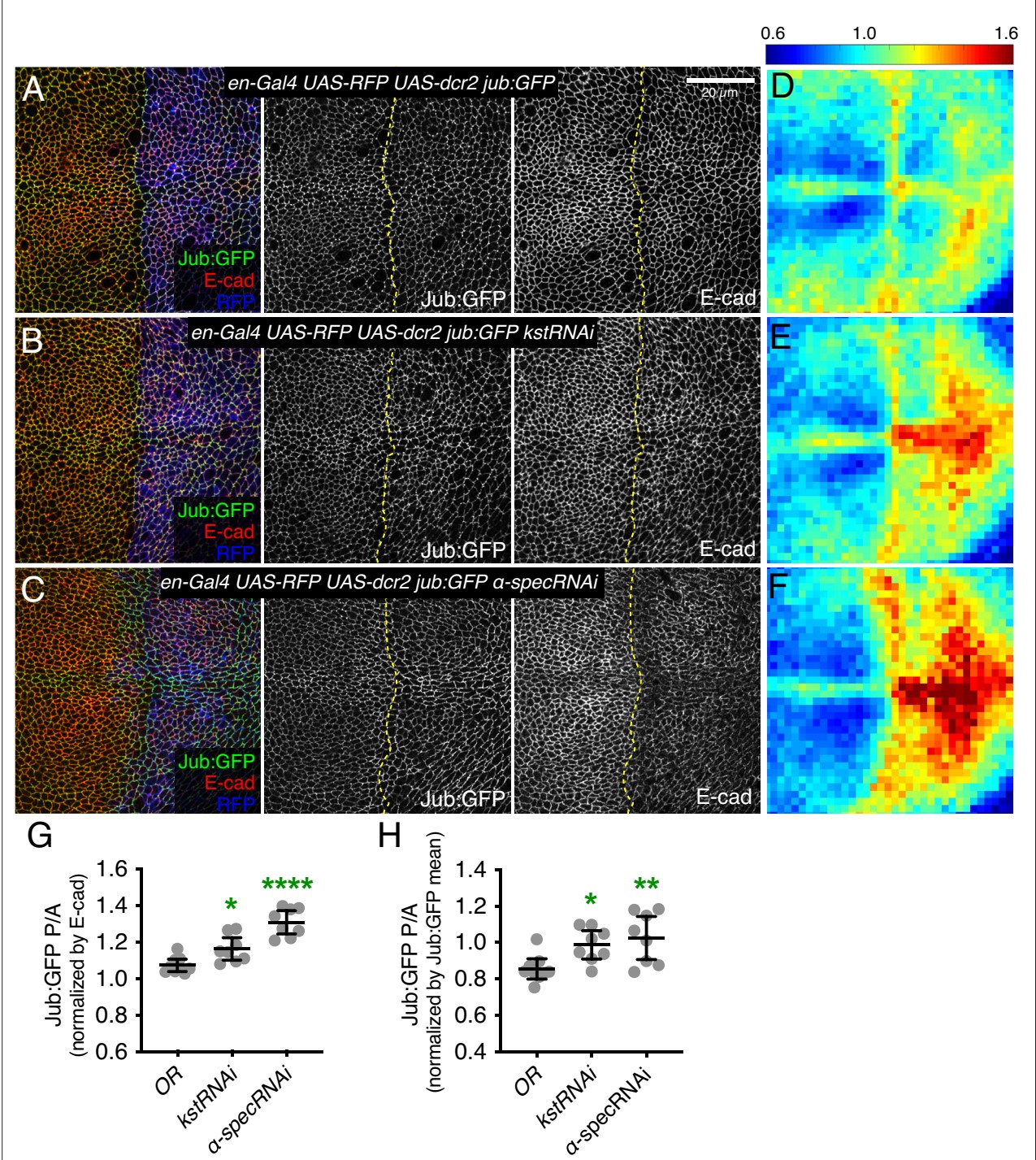

**Figure 2.** Knockdown of β-heavy spectrin (β$_H$-Spec) or α-spectrin (α-Spec) increases junctional Jub levels in wing discs. (**A–C**) Apical sections of wing discs expressing en-Gal4 UAS-RFP UAS-dcr2 Jub:GFP crossed to control (Oregon-R, OR) (**A**), UAS-kstRNAi(v37075) (**B**), or UAS-α-specRNAi (**C**) stained for E-cad (red/gray) showing the effect on Jub:GFP (green/gray) in the posterior compartment (marked by RFP, blue). Dashed yellow line marks the A/P compartment boundary. Panels to the right, in gray, show single channels, as indicated. Scale bar = 20 µm. (**D–F**) Heat maps of relative junctional Jub:GFP intensity of wing discs. Levels of Jub:GFP relative to E-cad levels are shown for the different genotypes analyzed. Heat map scale is indicated on the top. Number of wing discs used for analysis: Control (OR), n=9; UAS-kstRNAi, n=8; UAS-α-specRNAi, n=8. (**G–H**) Quantification of Jub:GFP normalized to E-cadherin (**G**) or to Jub:GFP mean intensity (**H**) in posterior cells compared to anterior cells (P/A) in wing discs expressing the indicated constructs, displayed as individual values. Data are shown as mean±95% CI. Statistical significance was determined by one-way analysis of variance (ANOVA) with Dunnett's multiple comparison test relative to the control (Oregon-R): ns: not significant, *p<0.05; **p≤0.01; ***p≤0.001; ****p≤0.0001.

*Figure 2 continued*

The online version of this article includes the following figure supplement(s) for figure 2:

**Figure supplement 1.** β-Heavy spectrin (β$_H$-Spec) modulates the recruitment of Jub:GFP to adherens junctions (AJs).

localization (*Figure 5C*). These observations suggest that while α-Spec needs β-Spec to localize to lateral membranes, the localization of β$_H$-Spec and α-Spec are functionally independent. To further examine the relationship between spectrins, we performed proximity ligation assays (PLA) using antibodies against α-Spec and GFP. A PLA signal consistent with close association of α-Spec and β-Spec was observed in wing discs expressing GFP-tagged β-Spec (*Figure 4E*). Conversely, no significant PLA signal was detected in wing discs expressing GFP-tagged β$_H$-Spec, implying that α-Spec and β$_H$-Spec are not closely associated in wing discs (*Figure 4D*). Together with the distinct localization of these proteins revealed by imaging, these observations argue against the existence of (αβ$_H$)$_2$ complexes in wing imaginal discs.

## β$_H$-Spec and apical myosin antagonize each other's localization to apical F-actin

The discovery that β$_H$-Spec and α-Spec localize independently led us to consider what factors might influence the apical localization of β$_H$-Spec. Intriguingly, the distribution of β$_H$-Spec in wing disc cells appears very similar to the apical distribution of F-actin and myosin, and similarities between the localization of β$_H$-Spec and myosin in *Drosophila* embryos have been noted previously (*Krueger et al., 2020*; *Thomas and Kiehart, 1994*). To determine whether myosin and β$_H$-Spec co-localize in wing discs, we imaged discs expressing GFP-tagged β$_H$-Spec (Kst:GFP) and mCherry-tagged myosin light chain (sqh-Sqh:mCherry). This revealed extensive co-localization between these proteins in the apical region of wing disc epithelial cells (*Figure 6A, B*). Quantitation of these images yielded a Pearson's correlation coefficient score of 0.636. For comparison, the Pearson's correlation coefficient score between Kst:YFP and α-Spec (*Figure 4A*) was 0.18. Both Kst:GFP and Sqh:mCherry also co-localized with F-actin, with Pearson's correlation coefficient scores of 0.641 and 0.612, respectively.

The extensive co-localization of β$_H$-Spec and myosin prompted us to investigate the functional relationship between them further. As noted above, reduction of β$_H$-Spec leads to increased apical myosin levels and activity. To investigate whether myosin reciprocally regulates β$_H$-Spec, we expressed transgenes that modulate actomyosin contractility. RNAi-mediated knockdown of Rok reduces myosin activity by reducing phosphorylation of myosin light chain (*Winter et al., 2001*). In wing discs, knockdown of *Rok* is also associated with reduced recruitment of myosin to apical junctions and reduced junctional tension (*Rauskolb et al., 2014*; *Figure 6—figure supplement 1C*), consistent with the generally positive correlation between myosin activity and co-localization with F-actin (*Fernandez-Gonzalez et al., 2009*; *Noll et al., 2017*). Conversely, examination of β$_H$-Spec in wing disc cells expressing *Rok* RNAi revealed increased levels of β$_H$-Spec along apical junctions (*Figure 6D*). To increase myosin activity, we expressed a constitutively activated, phosphomimetic form of myosin light chain, Sqh$^{EE}$ (*Winter et al., 2001*). This increases the recruitment of myosin to AJ and increases junctional tension (*Rauskolb et al., 2014*; *Figure 6—figure supplement 1D*), but decreases the recruitment of β$_H$-Spec to AJ, and a portion of β$_H$-Spec instead appears in apical vesicles (*Figure 6E, F*). To quantify these effects on β$_H$-Spec levels, we made maps of β$_H$-Spec intensity normalized against E-cad intensity, displayed on a red (high) to blue (low) heat map. Calculation of the ratio of intensities of the anterior (control) versus posterior (experimental) compartments further confirmed our observation that myosin antagonizes localization of β$_H$-Spec to AJ (*Figure 6G–J*). Thus, β$_H$-Spec and myosin II localization to AJ are affected in opposite ways by changes in cytoskeletal tension.

The opposing effects of changes in myosin activity on myosin and β$_H$-Spec localization, together with the observation that loss of β$_H$-Spec leads to increased myosin activity and apical localization, raised the possibility that myosin and β$_H$-Spec compete for localization to apical F-actin. To further investigate this possibility, we overexpressed β$_H$-Spec using a CRISPR-activator (CRISPRa) approach. This involves the expression of a transcriptional activator fused to dCas9 under UAS control, which can then be recruited to a gene of interest using a single-guide RNA (sgRNA) targeted upstream of the transcription start site (TSS) (*Jia et al., 2019*; *Jia et al., 2018*). To verify the overexpression of β$_H$-Spec, we employed it in flies with GFP-tagged β$_H$-Spec at the endogenous locus (Kst:GFP). To

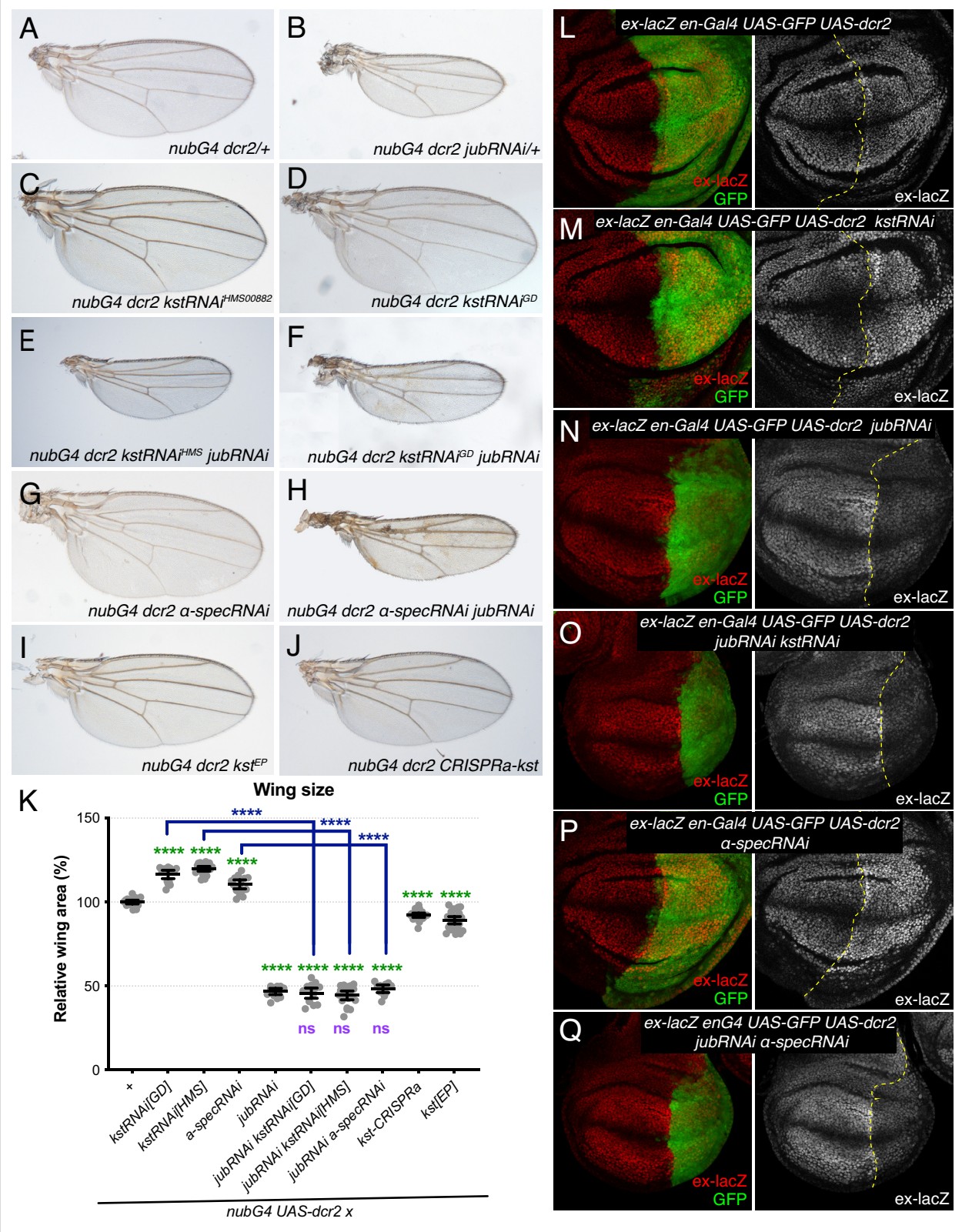

**Figure 3.** Jub is required for β-heavy spectrin (β_H-Spec) and α-spectrin (α-Spec) regulation of wing size and ex-lacZ. (**A–J**) Representative adult wings from flies cultured at 29°C and expressing UAS transgenes altering spectrin and/or Jub expression under control of a nub-Gal4 (nubG4) driver. (**K**) Quantification of wing area (mean±95% CI). Number of wing discs used for analysis: Control (OR), n=20; UAS-kstRNAi[GD], n=20; UAS-kstRNAi[HMS], n=21; UAS-α-specRNAi, n=20; UAS-jubRNAi, n=20, UAS-jubRNAi UAS-kstRNAi[GD], n=20; UAS-jubRNAi UAS-kstRNAi[HMS], n=20; UAS-

*Figure 3 continued*

jubRNAi UAS-α-specRNAi, n=20; UAS-kstCRISPRa, n=22; UAS-kst[EP], n=26. Statistical significance was determined by a one-way analysis of variance (ANOVA) with Tukey's multiple comparison test. Statistical comparisons are shown relative to nub-Gal4 UAS-dcr2/+in green, relative to nub-Gal4 UAS-dcr2 UAS-jubRNAi in purple and relative to UAS-kstRNAi in blue. (**L–Q**) Third-instar wing discs expressing ex-lacZ en-Gal4 UAS-dcr2 UAS-GFP (green) crossed to OR (**L**), UAS-kstRNAi (**M**), UAS-jubRNAi (**N**), UAS-jubRNAi UAS-kstRNAi (**O**), UAS-α-specRNAi (**P**), UAS-jubRNAi UAS-α-specRNAi (**Q**), stained for expression of ex-lacZ (red/white). Dashed yellow line indicates the A/P compartment boundary.

The online version of this article includes the following figure supplement(s) for figure 3:

**Figure supplement 1.** *kst* phenotypes are suppressed by reduction of tension.

avoid excessive cell death caused by Kst overexpression, we expressed this construct under inducible conditions, using the temperature-sensitive Gal4 repressor Gal80[ts]. Both imaging and western blotting of wing discs confirmed that this CRISPRa approach effectively increased expression of Kst:GFP (*Figure 7—figure supplement 1A–B*). Examination of myosin under β$_H$-Spec overexpression conditions, using Sqh:GFP, revealed a substantial reduction of myosin localization to AJs (*Figure 7A, C and E–F*), and a reduction in tension along cell junctions as assayed by the recoil speed after laser cutting (*Figure 7I*). Apical cell areas were also increased, consistent with a reduction of junctional tension (*Figure 7B* and *Figure 7—figure supplement 1*). As an independent method to overexpress β$_H$-Spec, we used a previously described EP-element insertion near *kst*, P[EPgy2]EY01010 (UAS-kst[EP]) (*Pogodalla et al., 2021*). Overexpression of β$_H$-Spec using UAS-kst[EP] under *en-Gal4* control also reduced myosin levels at AJ (*Figure 7—figure supplement 1C*), although the effect appeared weaker than that induced by the CRISPRa approach. These results provide further evidence that β$_H$-Spec antagonizes myosin recruitment to AJs and establish that this effect can be observed under both increased and decreased β$_H$-Spec expression conditions.

Consistent with these reductions in apical myosin accumulation, β$_H$-Spec overexpression also reduced junctional recruitment of Jub (*Figure 7B, D, G, and H* and *Figure 7—figure supplement 1D*) and decreased wing size (wings were 92% and 89% of *nub-Gal4 UAS-dcr2* control size, for the two different overexpression constructs) (*Figure 3I–K*).

## β$_H$-Spec and myosin compete for binding to F-actin

The reciprocal antagonism between myosin and β$_H$-Spec suggested that they could compete for binding to F-actin. β$_H$-Spec contains two N-terminal actin-binding domains, CH1 and CH2 (*Liem, 2016*), and myosin contains an actin-binding region in its motor domain (*Duan et al., 2018*). While spectrins and myosin have been purified and characterized *in vitro*, we lack a mechanistic understanding of the interaction between β$_H$-Spec and F-actin and how it might affect myosin binding. To address this, we conducted *in vitro* co-sedimentation assays with purified protein domains and F-actin. We found that the isolated *Drosophila* β$_H$-Spec actin-binding region binds F-actin weakly (*Figure 8A*), in agreement with previous reports of F-actin binding by other spectrin CH1-CH2 domains (*Avery et al., 2017*; *Duan et al., 2018*). Constitutively active *Drosophila* myosin II (subfragment-1-like protein) binding to F-actin appears stronger than β$_H$-Spec binding, as at the same concentration a greater fraction of myosin II is bound to F-actin (*Figure 8A, B*). To investigate the antagonism between these proteins observed *in vivo*, we performed biochemical competition assays between the β$_H$-Spec CH domains and myosin for binding to F-actin. F-actin was preincubated with an excess of β$_H$-Spec CH domains to maximize binding to F-actin, and then increasing concentrations of myosin were added. We found that myosin can displace the actin-binding region of β$_H$-Spec from F-actin by ~50% at the highest myosin concentration that we could use (*Figure 8C*, bottom right). Myosin binding to F-actin was unaffected by preincubation with β$_H$-Spec CH domains (*Figure 8—figure supplement 1A*), likely due to active myosin's higher binding affinity for F-actin.

To gain a structural understanding of the antagonism between β$_H$-Spec and myosin, we built homology models of the *Drosophila* CH1-CH2 domain of β$_H$-Spec and the myosin II motor domain and compared them with previous cryo-EM structures of the human actin-bound β-III spectrin actin binding CH1 domain and actin-bound myosin (*Avery et al., 2017*; *von der Ecken et al., 2016*). The superimposition of the isolated *Drosophila* β$_H$-Spec CH1 domain model on the human β-III spectrin CH1 domain in a complex with F-actin structure revealed a binding site between actin subdomains (SD) SD1 and SD2 along the filament (*Figure 8D*). The binding region of myosin on F-actin includes SD1, SD2, and SD3 (*Figure 8D'*). The superimposition of both models suggests that the binding position of

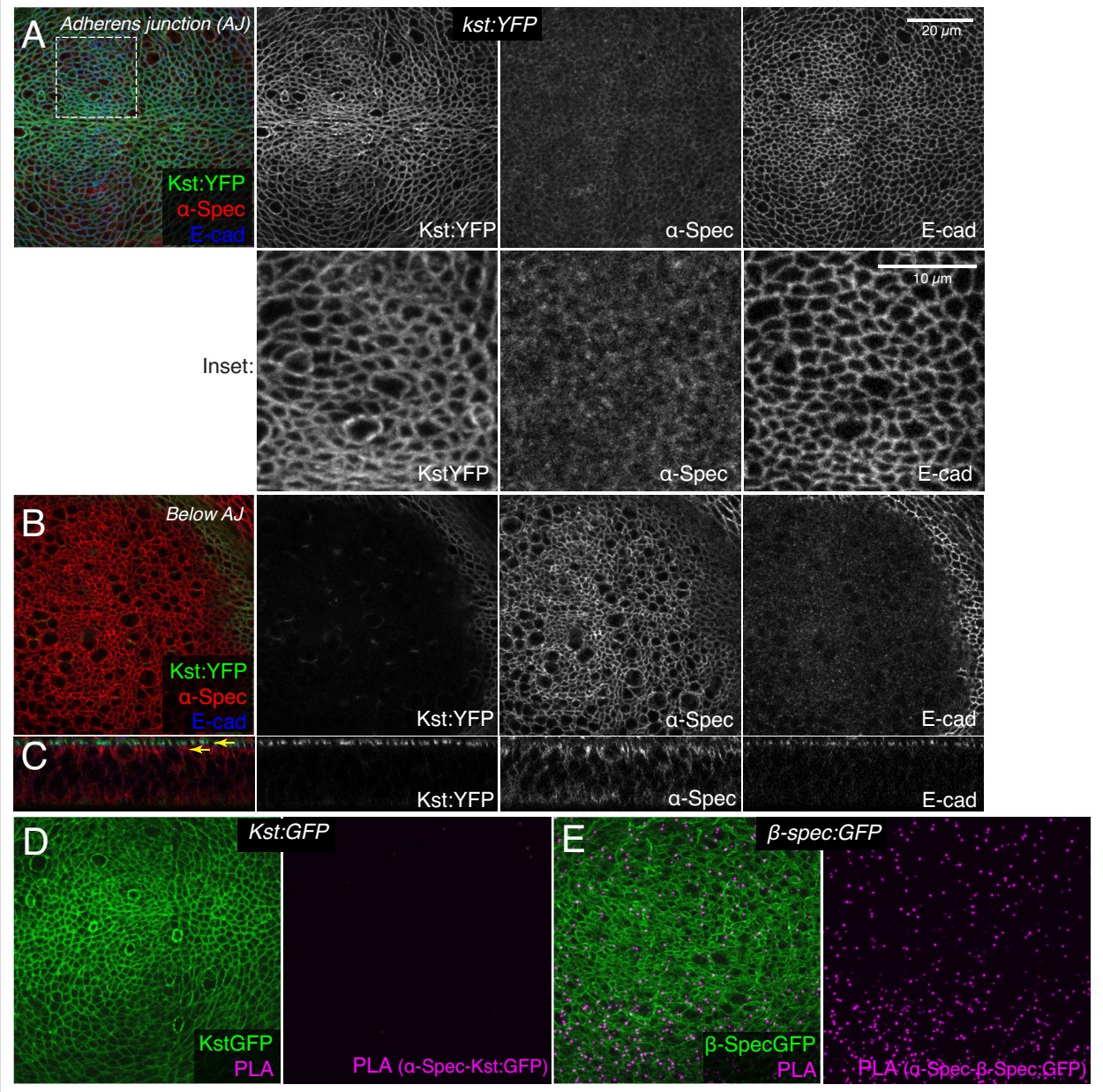

**Figure 4.** β-Heavy spectrin (β$_H$-Spec) and α-spectrin (α-Spec) do not co-localize in wing discs. (A–C) Wing discs expressing Kst:YFP immunostained with α-Spec (red/gray) and E-cad (blue/gray) antibodies showing the localization of β$_H$-Spec (Kst:YFP, green/gray) and α-Spec at the adherens junction (AJ) (**A**), below the AJ (**B**) and in cross sections (**C**). Upper yellow arrow in cross section indicates AJ layer, lower yellow arrow indicates 'Below AJ' layer. Inset shows higher magnification of single channels from the boxed region in A. (D, E) Wing discs expressing Kst:GFP (**D**) or β-Spec:GFP (**E**), with GFP in green and signal from proximity ligation assays (PLA) using rabbit anti-GFP and mouse α-Spec antibodies in magenta.

the CH1 domain on F-actin sterically interferes with the formation of a strong actin-myosin interface (**Figure 8D''**). Modeling of the β$_H$-Spec CH1-CH2 domain indicates that the CH2 domain presents additional steric hindrance for myosin binding to F-actin (**Figure 8—figure supplement 1B–B''**). Thus, structural modeling suggests that the binding of spectrin or myosin to individual binding sites on F-actin is mutually exclusive, which could explain why they compete for F-actin association *in vivo* and *in vitro*.

## Discussion

Multiple models for how spectrins regulate Hippo signaling have been proposed (**Deng et al., 2015**; **Deng et al., 2020**; **Fletcher et al., 2015**; **Wong et al., 2015**). We focused on investigating claims that

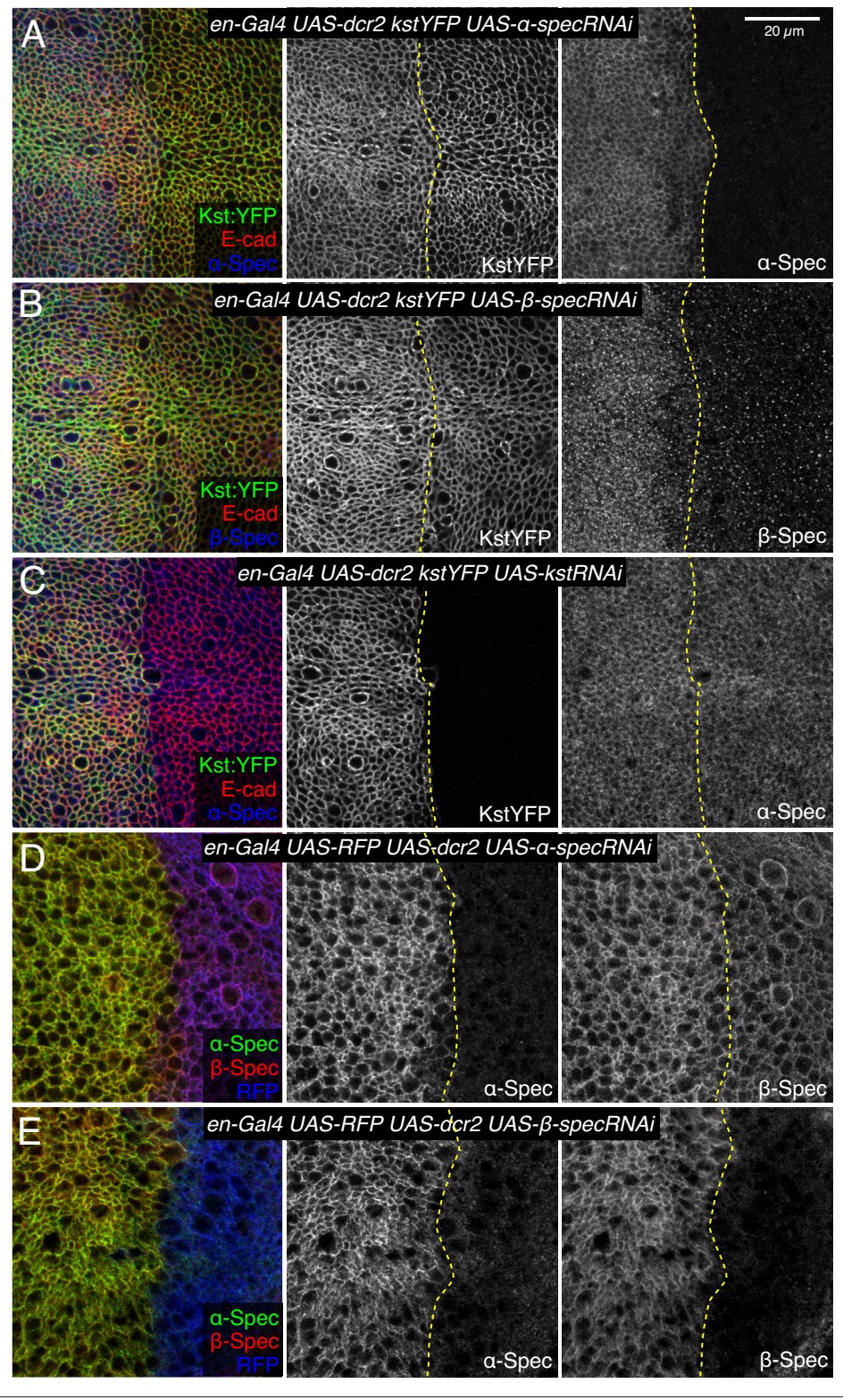

**Figure 5.** Influence of spectrin knockdowns on spectrin subunit localization in wing discs. (**A–C**) Apical sections of wing discs expressing en-Gal4 UAS-dcr2 Kst:YFP (green/gray) along with UAS-α-specRNAi (**A**), UAS-β-specRNAi (**B**), and UAS-kstRNAi (**C**) immunostained for α-spectrin (α-Spec) (A and C, blue/gray) or β-spectrin (β-Spec) antibodies (B, blue/gray). The adherens junction (AJ) layers were obtained by ImSAnE, using as a reference the E-cad channel and seven layers were projected (2.1 μm). (**D,E**) Sections of wing discs expressing en-Gal4 UAS-RFP

*Figure 5 continued on next page*

*Figure 5 continued*

(blue) UAS-dcr2 crossed with UAS-α-specRNAi (**D**) or UAS-β-specRNAi (**E**) stained with mouse α-Spec (green/gray) and rabbit β-Spec antibody (red/gray). Dashed yellow lines indicate the A/P compartment boundary. Scale bar = 20 μm.

spectrins could alter cytoskeletal tension in wing discs by regulating pMLC levels without affecting localization of myosin or Jub (*Deng et al., 2015*). In contrast to prior studies, we observed that when β_H-Spec or α-Spec levels are decreased by RNAi, levels of junctional myosin are increased. This increase in junctional myosin levels is associated with increased junctional tension and with increased recruitment of Jub to AJ. We are not certain why these effects were missed in prior studies, but we note that our observations are consistent with studies linking recruitment of both myosin and Jub to AJ under tension (*Alégot et al., 2019*; *Fernandez-Gonzalez et al., 2009*; *Ibar et al., 2018*; *Noll et al., 2017*; *Rauskolb et al., 2019*; *Rauskolb et al., 2014*; *Razzell et al., 2018*; *Sarpal et al., 2019*). Moreover, our results are further supported by the observation that *jub* is genetically required for the influence of spectrin knockdown on Yki activity and wing growth. Taken together, these observations imply that β_H-Spec and α-Spec regulate Hippo signaling in wing discs through the Jub biomechanical pathway, rather than through hypothesized alternate mechanisms.

Spectrin has been suggested to form two distinct complexes in *Drosophila* epithelial cells, $(\alpha\beta)_2$ and $(\alpha\beta_H)_2$ heterotetramers, which localize to the lateral and apical sides of cells, respectively (*Deng et al., 2015*; *Dubreuil et al., 1997*; *Fletcher et al., 2015*; *Lee et al., 1997*; *Thomas et al., 1998*; *Zarnescu and Thomas, 1999*). However, our observations indicate that β_H-Spec functions independently of α-Spec in wing imaginal discs. α-Spec and β_H-Spec do not exhibit significant co-localization. Moreover, α-Spec and β_H-Spec are not required for each other's localization to apical cell junctions. This contrasts with the requirement for β-Spec for recruitment of α-Spec to lateral membranes in wing discs. The requirement is not entirely reciprocal, as α-Spec knockdown only partially reduces β-Spec recruitment, but this likely reflects mechanisms that recruit spectrins to cell membranes: β-Spec subunits, but not α-Spec subunits, have a pleckstrin homology domain that can mediate membrane association, as well as possessing the CH domains that mediate F-actin association. Thus β-Spec can associate with lateral membranes without α-Spec, but α-Spec does not have a way to associate with lateral membranes without β-Spec.

An independent role for β_H-Spec has also been suggested in mammalian photoreceptors, where it was reported that that mammalian βV-Spec does not co-localize with αII-spectrin, and that βV-Spec could form homodimers, potentially allowing it to cross-link actin by itself, enabling α-Spec-independent functions (*Papal et al., 2013*). The observation that a mutation in *Drosophila* α-Spec that disrupts binding to β-Spec *in vitro* has only mild phenotypes also suggests that spectrin functions do not depend entirely on αβ interactions (*Khanna et al., 2015*). Collectively, our results together with these earlier studies emphasize that the dogma that "spectrin comprises α- and β-subunits that interact in an antiparallel manner to form an αβ dimer" (*Liem, 2016*) should be revised. Nonetheless, our results do not exclude the possibility that β_H-Spec and α-Spec might act together in a physical complex in other tissues, or under distinct physiological conditions.

The conclusion that β_H-Spec and α-Spec act independently in wing discs implies that they influence tension at apical junctions through distinct mechanisms. We suggest that α-Spec could influence AJ tension in wing discs through the mechanism proposed to explain the influence of α- and β-Spec in pupal eyes (*Deng et al., 2020*). It was inferred that α- and β-Spec maintain cell rigidity by linking F-actin to membranes. In the absence of α- or β-Spec, it was proposed that dissociation between F-actin and the membrane leads to an expansion of the apical regions. This expansion increases cytoskeletal tension at AJs, which bind F-actin independently of spectrins. Consistent with this suggested mechanism, we observed a decrease in cell height in α-Spec knockdown cells in wing discs, in conjunction with increased tension at apical junctions. The alteration in cell shape was not observed in β_H-Spec knockdown cells, further supporting the conclusion that β_H-Spec and α-Spec act in different ways to regulate junctional tension.

Instead, our experiments analyzing the relationship between β_H-Spec and myosin revealed an entirely different mechanism by which β_H-Spec influences tension at AJ. We observed a mutual antagonism between β_H-Spec and myosin *in vivo* for localization to apical F-actin: decreasing β_H-Spec increases junctional myosin, while increasing β_H-Spec decreases junctional myosin. Reciprocally,

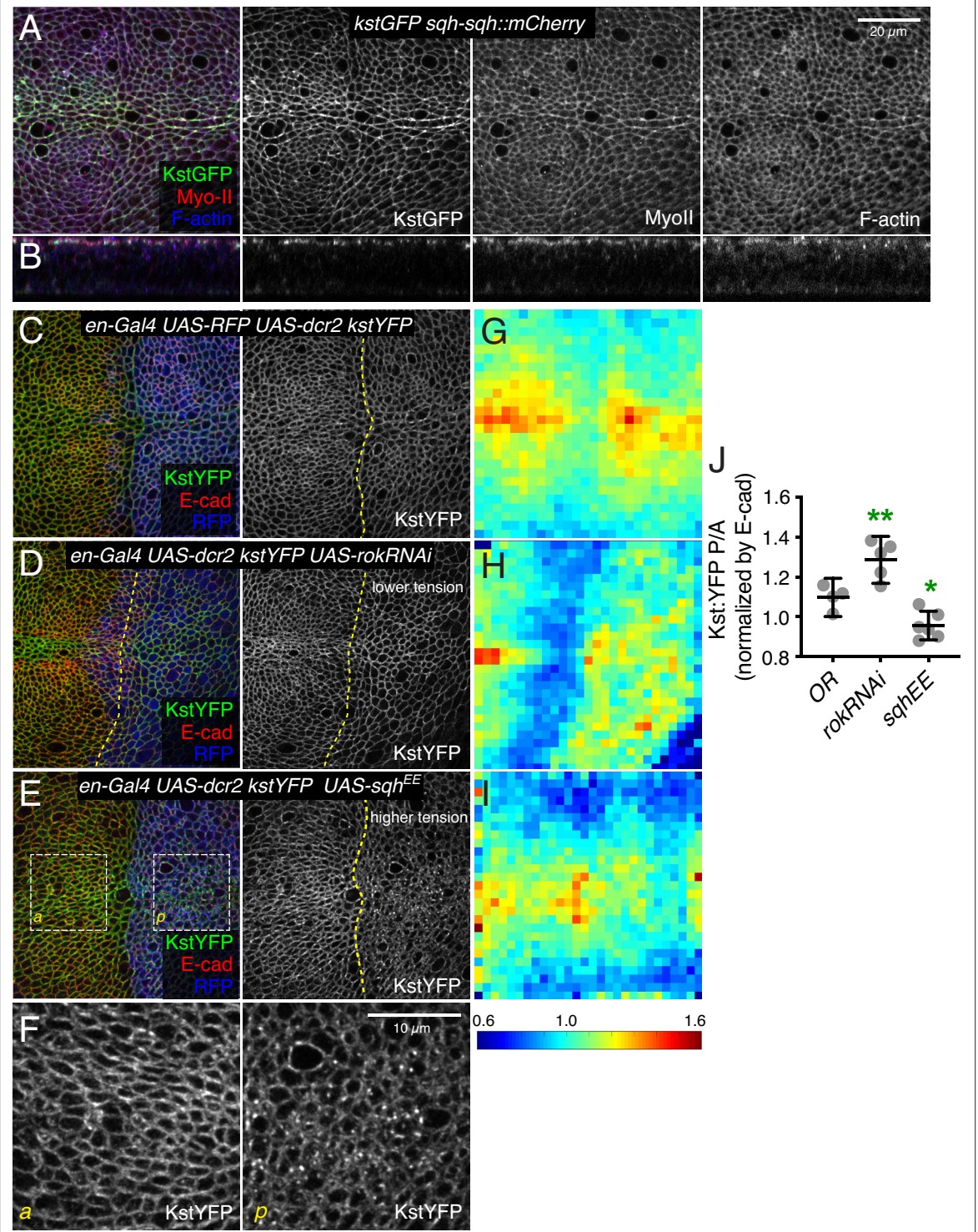

**Figure 6.** β-Heavy spectrin (β_H-Spec) co-localizes with myosin and is regulated by myosin activity. (**A,B**) Wing disc co-expressing Kst:GFP (green/gray) and the labeled myosin II subunit sqh:mCherry (red/gray) in apical horizontal sections (**A**) and lateral sections (**B**), co-stained with phalloidin for F-actin (blue/gray). (**C–E**) Apical sections of wing discs expressing en-Gal4 UAS-RFP UAS-dcr2 kst:YFP crossed to (Oregon R, OR) (**C**), UAS-rokRNAi (**D**), or UAS-sqhEE (**E**) in the posterior compartment (marked by RFP, blue), showing the effect of altering myosin activity on Kst:YFP. Dashed yellow lines mark the A/P compartment boundary. Scale bar = 20 μm. (**F**) Higher magnification of the boxed regions indicated in E. Scale bar = 10 μm. (**G–I**) Heat maps of junctional Kst:YFP intensity relative to E-cad in en-Gal4 UAS-RFP UAS-dcr2 kst:YFP crossed to (Oregon R, OR) (**C**), UAS-rokRNAi (**D**), or UAS-sqhEE (**E**) wing discs (n=5 for each genotype), as in the representative examples shown in C–E. Heat map scale is at bottom. (**J**) Quantification of Kst:YFP intensity normalized to E-cadherin in posterior cells (P) compared to anterior cells (A) in wing discs expressing the indicated constructs under en-Gal4

*Figure 6 continued on next page*

*Figure 6 continued*

control (n=5). Data are shown as mean±95% CI, error bars indicate CI. Statistical significance was determined by a one-way analysis of variance (ANOVA) with Dunnett's multiple comparison test relative to the control (Oregon-R): ns: not significant, *p<0.05; **p≤0.01.

The online version of this article includes the following figure supplement(s) for figure 6:

**Figure supplement 1.** Additional analysis of β-heavy spectrin (β$_H$-Spec) localization and the effect of changes in tension on myosin.

increasing myosin activity decreases β$_H$-Spec localization to apical F-actin, while decreasing myosin activity increases β$_H$-Spec localization to apical F-actin. *In vitro* studies with purified protein domains revealed that myosin can compete with β$_H$-Spec for binding to F-actin. Finally, computational modeling of protein structures revealed that myosin and β$_H$-Spec would interfere with each other's binding to F-actin. Together, these observations indicate that β$_H$-Spec and myosin can directly compete with each other for localization to F-actin. We suggest therefore that the influence of β$_H$-Spec on junctional tension is likely to be a direct consequence of its competition with myosin for overlapping binding sites on F-actin.

Despite β$_H$-Spec and myosin sharing overlapping binding sites on F-actin, and competing reciprocally *in vivo*, in our co-sedimentation experiments we could only detect a partial ability of myosin to compete for β$_H$-Spec binding, and we could not detect an ability of β$_H$-Spec to compete for myosin binding. Several factors are likely to contribute to these observations. First, we could not use higher protein concentrations of myosin or β$_H$-Spec due to the need to keep the salt concentration constant and close to physiological levels, and to prevent protein precipitation. Second, it has been suggested for β-Spec that the CH2 domain regulates the actin binding function of the CH1 domain through steric hindrance when the two domains are associated (*Avery et al., 2017*). A specific mutation in β-Spec CH2 (L253P) has been shown to lower the energetic barrier between closed and open structural states, increasing the affinity of β-Spec for F-actin around 1000-fold (*Avery et al., 2016*), but it is unknown how β-Spec or β$_H$-Spec conformational changes are normally regulated *in vivo* and whether both proteins share this regulatory feature. Additionally, phosphorylation of myosin regulatory light chain shifts myosin from a compact, autoinhibited conformation to a filamentous, active conformation (*Kiehart and Feghali, 1986*; *Vasquez et al., 2016*). The autoinhibited conformation binds F-actin very weakly ($K_D$>100 µM) (*Heissler and Manstein, 2013*; *Sellers et al., 1982*) compared to the active conformation, suggesting that spectrin could outcompete autoinhibited myosin more effectively than active myosin for binding to F-actin. Our *in vitro* experiments used an active form of myosin. In addition, other factors including other actin-binding proteins and cytoskeletal tension are likely to influence the dynamic localization and actin-binding properties of both proteins (*Duan et al., 2018*; *Greenberg et al., 2016*).

The competition between β$_H$-Spec and myosin also provides key insights into how β$_H$-Spec contributes to ratcheting of apical constriction. The apical constriction of cells in the ventral furrow that initiates *Drosophila* mesoderm invagination occurs through fast constriction pulses interrupted by pauses during which cells must stabilize their constricted state before reinitiating constriction (*Martin et al., 2009*; *Xie and Martin, 2015*). This ratcheting-like behavior is thought to be a consequence of the finite length of actin filaments. Myosin contracts the cytoskeleton by driving filaments past each other, and extensive contractions require release and reassociation with new pairs of filaments. β$_H$-Spec participates in ratcheting of apical constriction (*Krueger et al., 2020*). When β$_H$-Spec is knocked down, cells can undergo cycles of unratcheted apical constriction during which they alternately constrict and then expand. Consequently, most β$_H$-Spec-depleted embryos fail to complete normal mesoderm invagination. It was proposed that the actin cross-linking function of β$_H$-Spec could hold F-actin in place for the next cycle of myosin-mediated contraction, but this raises the question of how myosin and β$_H$-Spec association with F-actin are coordinated so that β$_H$-Spec prevents relaxation without interfering with constriction. Our results suggest a simple solution: since they compete for the same binding site, the release of myosin from F-actin at the end of a cycle of contraction would naturally be coupled to the accessibility of F-actin for binding by β$_H$-Spec. Thus, the competition between myosin and β$_H$-Spec for binding to F-actin enables myosin-mediated cell contraction to effectively alternate with β$_H$-Spec-mediated stabilization.

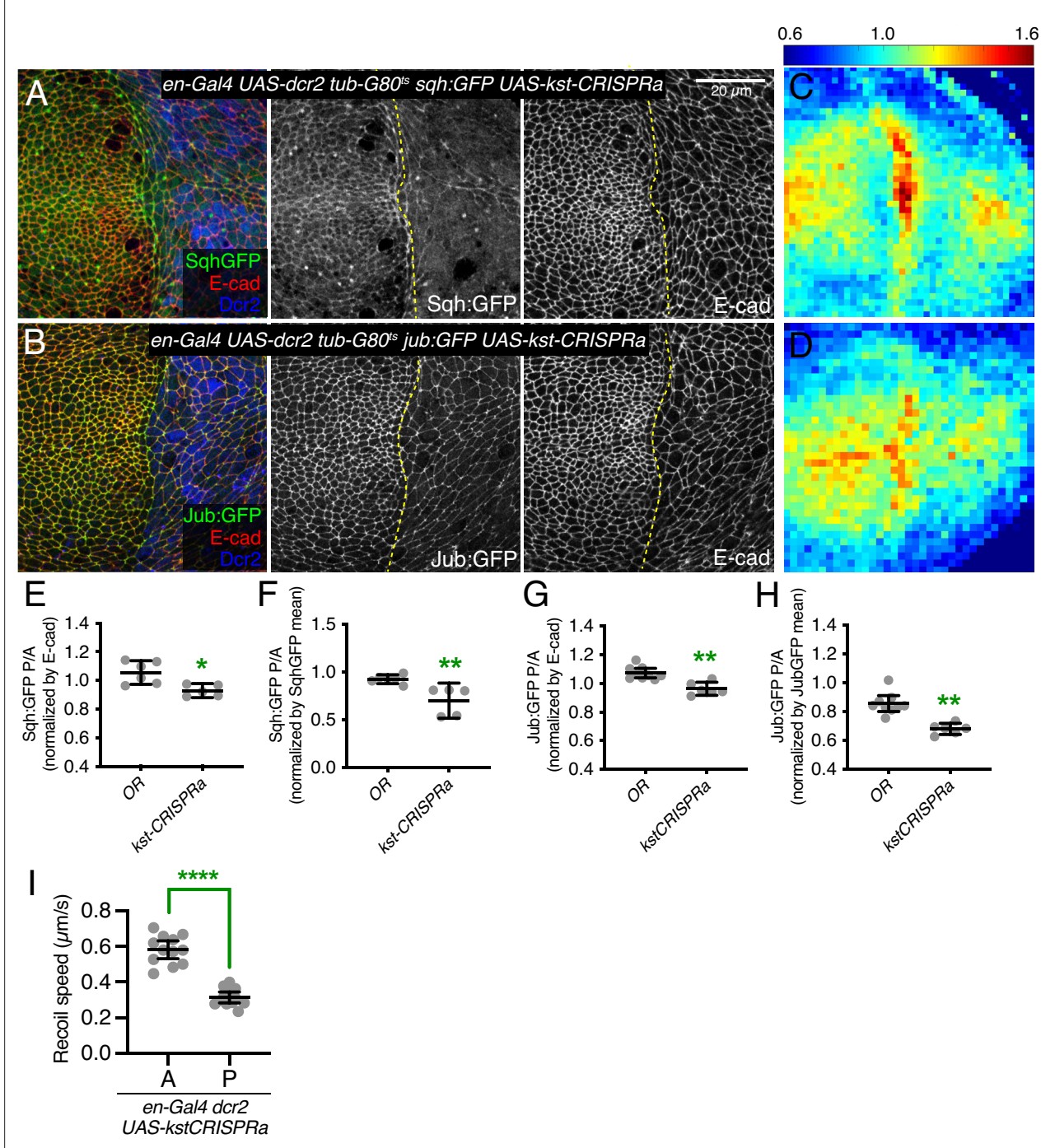

**Figure 7.** β-Heavy spectrin (β$_H$-Spec) overexpression reduces in junctional tension in wing discs. (**A–B**) Apical sections of wing imaginal discs expressing en-Gal4 UAS-RFP UAS-dcr2 sqh:GFP (**A**) or en-Gal4 UAS-RFP UAS-dcr2 jub:GFP (**B**) crossed to UAS-kst-CRISPRa showing the effect on Sqh:GFP or Jub:GFP levels and localization in the posterior compartment (marked by RFP, blue). Yellow dashed line indicates the A/P compartment boundary. Scale bar = 20 µm; all images are at the same magnification. (**C–D**) Heat maps of relative junctional Sqh:GFP (**C**) or Jub:GFP (**D**) intensity of wing discs. Levels of Sqh:GFP relative to E-cad levels are shown for the different genotypes analyzed in A–B. Heat map scale is indicated on the top. Number of wing discs used for analysis: UAS-kst-CRISPRa with Sqh:GFP, n=5; UAS-kst-CRISPRa with Jub:GFP, n=6. (**E–F**) Quantification of Sqh:GFP overlapping E-cad in posterior cells (**P**) compared to anterior cells (**A**) in wing disc expressing the indicated constructs, displayed as individual values, normalized by E-cad (**E**) or normalized by the mean intensity of Sqh:GFP (**F**). (**G–H**) Quantification of Jub:GFP overlapping E-cad in posterior cells (P) compared to anterior cells (A) in wing disc expressing the indicated constructs, displayed as individual values, normalized by E-cad (**G**) or normalized by the mean intensity of Jub:GFP (**H**). (**I**) Average recoil velocities after laser cutting of cell junctions in anterior (**A**) or posterior (**P**) compartments of wing discs expressing

*Figure 7 continued on next page*

*Figure 7 continued*

UAS-kstCRISPRa in posterior cells (n=20). Data are shown as mean±95% CI. Statistical significance for (**E-I**) was determined by Student's t-test, *p<0.05; **p≤0.01; ***p≤0.001; ****p≤0.0001.

The online version of this article includes the following figure supplement(s) for figure 7:

**Figure supplement 1.** Additional analysis of βH-Spec overexpression.

## Materials and methods

### *Drosophila* genetics

Unless otherwise indicated, crosses were performed at 29°C. Protein localization and expression levels were monitored using previously characterized transgenes: *ex-lacZ* (**Hamaratoglu et al., 2006**), *kst:YFP* (**Lye et al., 2014**), *kst:GFP* (**Nagarkar-Jaiswal et al., 2015**), *jub:GFP* (**Sabino et al., 2011**), *sqh:GFP* (**Royou et al., 2004**), *sqh:mCherry* (**Martin et al., 2009**), and *β-Spec:GFP* (II) (this paper).

To manipulate gene expression in the posterior compartment, *en-Gal4 UAS-RFP; UAS-dcr2* flies were crossed with to UAS-RNAi or overexpression lines. RNAi transgenes used were *UAS-kstRNAi* (v37075), *UAS-kstRNAi* (HMS00882), *UAS-α-specRNAi* (v25387), *UAS-β-specRNAi* (GL01174), *UAS-rokRNAi* (v104675), and *UAS-jubRNAi* (v38442). To increase myosin activity, we used *UAS-sqh^{EE}* (**Winter et al., 2001**), and to increase $\beta_H$-Spec levels, we used *UAS-kstCRISPRa* (II) (this paper) and *UAS-kst^{P[EPgy2]EY01010}* (**Pogodalla et al., 2021**).

### DNA cloning

To overexpress $\beta_H$-Spec (*UAS-kstCRISPRa*), we used a second-generation CRISPR/Cas9-transcriptional activation approach (**Jia et al., 2018**), allowing us to recruit the transcriptional machinery near the TSS of *kst* under UAS control. For this, we made the following primers to generate a gRNA located less than 400 nt from the TSS: 5'-TTCGGATAAGCCGACAGGGTCTAT and 5'-AAACATAGACCCTGTC GGCTTATC-3'. These primers were duplexed and cloned in the FlySAM 2.0 vector using the BbsI site (**Jia et al., 2019**). Transgenic flies were made by injection, inserting the construct into the attP40 site (BestGene).

The actin binding domain (aa 1–278) of *kst* was cloned into pGEX-3X at the EcoRI site, using the following primers: 5'-gatctgatcgaaggtcgtggaATGACCCAGCGGGACGGC-3' and 5'-atcgtcag tcagtcacgatgTTACTTCTTGCGATCTGCGTCCATTAGC-3', and assembled using NEBuilder HiFi DNA Assembly (New England Biolabs, E2621) to generate plasmid pGEX-3X-ABDkst.

Recombineering was used to generate the β-Spec:GFP construct. A left homology arm (LHA, 1 kb before the stop codon of β-Spec) and a right homology arm (RHA, 1 kb after the stop codon of β-Spec) were cloned in pL452-cEGFP (**Venken et al., 2008**) by Gibson assembly into EcoRI (for LHA) and NotI (for RHA) sites, respectively, using the following primers: LHA_b-spec_FWD 5' gacctgcagcca agctatcgATACATGGCTGCCAAGGC 3'; LHA_b-spec_REV 5' gatcggaattgggctgcaggCTTTTTCTTTAA AGTAAAAAACGATCTGC 3'; RHA_b-spec_FWD 5' caagtaactagttctagagcAGTAACAGCCGTAACG CAAC 3' and RHA_b-spec_FWD 5' tggagctccaccgcggtggcCGGCAATTGGTGTACTTTAAAG 3'. The regions of the primers in lowercase indicate homology with the vector.To activate the recombination machinery, SW106 *Escherichia coli* containing the P[acman] clone CH322-20K3 (chloramphenicol resistant) (**Venken et al., 2009**) were incubated at 42°C for 17 min before making electrocompetent cells. A fragment containing LHA-loxP-NeoR/KanR-loxP-EGFP-RHA was amplified and electroporated into the cells. Recombinant clones were selected by chloramphenicol/kanamycin resistance and then floxed by inducible Cre expression by adding 0.1% L-arabinose. The final construct (LHA-loxP-EGFP-RHA) was confirmed by Sanger sequencing and injected for insertion in flies into the attP40 site (BestGene).

### Histology and imaging

For most experiments wing discs were fixed in 4% paraformaldehyde for 15 min at room temperature. Sqh:GFP discs were fixed for 12 min. Primary antibodies used were mouse anti-α-Spec (1:50, Developmental Studies Hybridoma Bank [DSHB], 3A9, deposited to the DSHB by Branton, D/Dubreuil, R, RRID:AB_528473), rabbit anti-β-Spec (1:100, a gift from Christian Klämbt) (**Hülsmeier et al., 2007**), rabbit anti-Dcr2 (1:800, Abcam, ab4732, RRID:AB_449344), rabbit anti-pMLC (T18/S19) (1:50, Cell Signaling Technologies, #3671, RRID:AB_330248), rat anti-E-cad (1:200, DSHB, DCAD2-c, deposited to the DSHB by Uemura, T, RRID:AB_528120), mouse β-galactosidase (1:200,

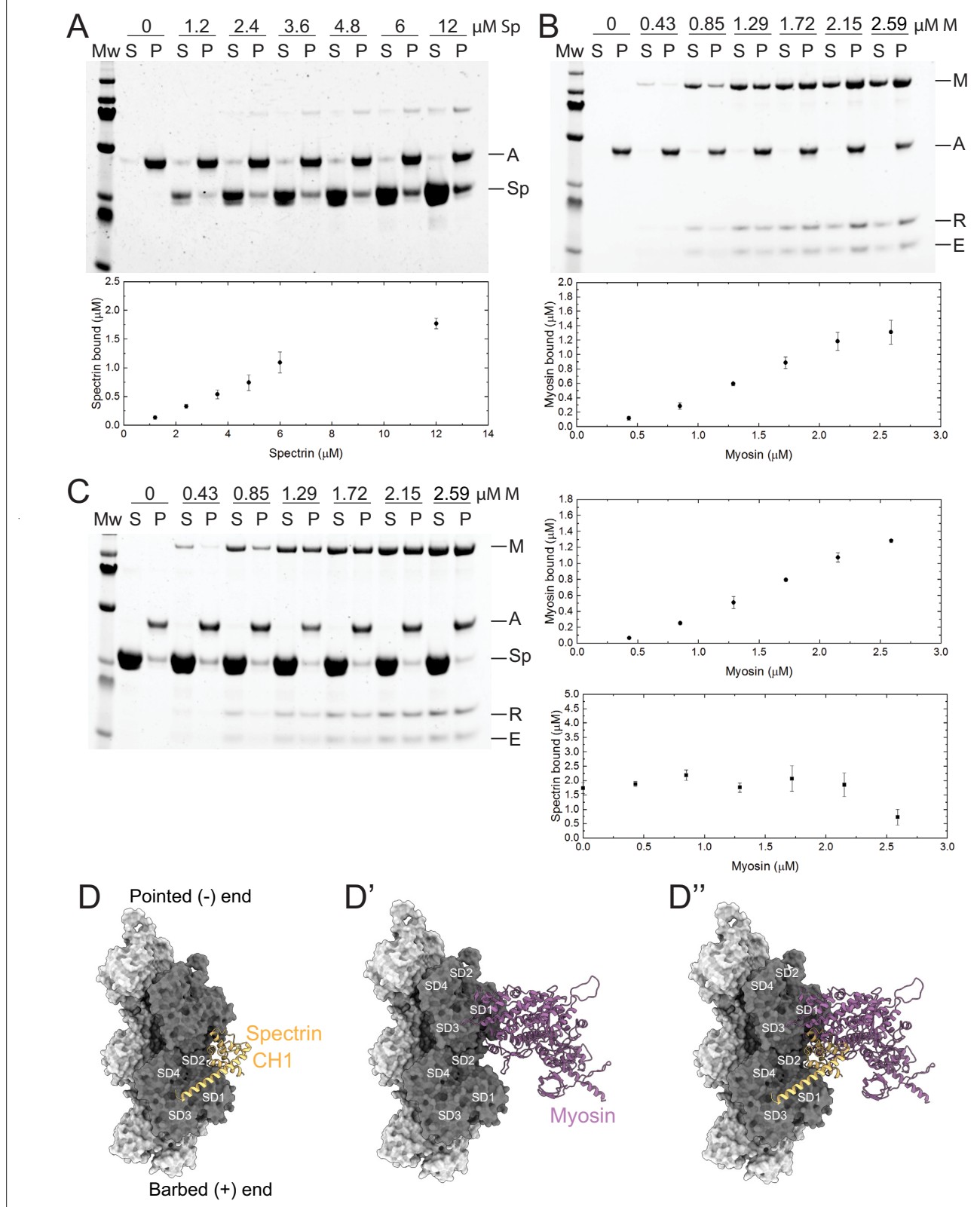

**Figure 8.** β-Heavy spectrin (β_H-Spec) and myosin have overlapping binding sites on F-actin. (**A–C**) Co-sedimentation assays with β_H-Spec CH domains (spectrin), F-actin, and myosin II subfragment-1-like protein (myosin). For all sedimentation assays, A, M, Sp, E, and R refer to F-actin, myosin heavy chain, spectrin, myosin essential light chain, and myosin regulatory light chain, respectively. Mw indicates marker and S and P refer to supernatant and pellet, respectively. Quantification shows mean ± SD of the sedimentation behavior of spectrin and myosin with F-actin (n=3). (**A**) Co-sedimentation

*Figure 8 continued on next page*

*Figure 8 continued*

assay between spectrin (0–12 µM) and F-actin (2 µM). (**B**) Co-sedimentation assay between myosin (0–2.59 µM) and F-actin (2 µM). (**C**) Co-sedimentation assay between spectrin (12 µM), myosin (0–2.59 µM), and F-actin (2 µM). Quantification shows that myosin reduces spectrin binding by ~50% (bottom panel). (**D–D''**) Model of the *Drosophila* CH1 domain (yellow) bound to F-actin (gray). (**D**) Model of the *Drosophila* myosin motor domain (purple) bound to F-actin (gray). (**D'**) Superimposition of the CH1 and myosin motor domain on F-actin. The two strands of the actin filament are shown in light and dark gray and subdomains are indicated.

The online version of this article includes the following figure supplement(s) for figure 8:

**Figure supplement 1.** Additional examination of competition between β-heavy spectrin (β$_H$-Spec) and myosin for F-actin.

DSHB, JIE7, deposited to the DSHB by Mason, TL/Partaledis, JA, RRID:AB_528101). Secondary antibodies were used at a 1:100 dilution, and included anti-rat Alexa Fluor 647 (Jackson ImmunoResearch, 712-605-153, RRID:AB_2340694), anti-rabbit Alexa Flour 647 (Jackson ImmunoResearch, 711-605-152, RRID:AB_2492288), anti-mouse Alexa Fluor 647 (Jackson ImmunoResearch, 715-605-151, RRID:AB_2340863), anti-mouse Cy3 (Jackson ImmunoResearch, 715-165-151, RRID:AB_2315777), anti-rabbit Cy3 (Jackson ImmunoResearch, 711-165-152, RRID:AB_2307443), anti-rat Cy3 (Jackson ImmunoResearch, 712-165-153, RRID:AB_2340667), anti-mouse Alexa Fluor 488 (Thermo Fisher Scientific, A-21202, RRID:AB_141607) and anti-rabbit Alexa Flour 488 (Thermo Fisher Scientific, A-21206, RRID:AB_2535792). DNA was stained using Hoechst (Invitrogen, H3570). Wing discs were removed and mounted on a slide in Vectashield (Vector Laboratories, H-1000). Confocal images were captured on a Leica SP8 microscope.

## Proximity ligation assay

PLA was performed with the Duolink Proximity Ligation Assay kit according to the manufacturer's instructions (Sigma). Fixation was performed as in the normal immunostaining procedure. For the permeabilization step, 0.5% Triton X-100 in PBS was used in two washes of 20 min each. Antibodies used for PLA include rabbit anti-GFP (1:200; ChromoTek # PABG1-20, RRID:AB_2749857) and mouse α-Spec (1:50, DSHB, 3A9, RRID:AB_528473) antibodies, and secondary anti-mouse MINUS (Sigma-Aldrich Cat# DUO92004, RRID:AB_2713942) and anti-rabbit PLUS (Sigma-Aldrich Cat# DUO92002, RRID:AB_2810940) probes were used.

## Immunoblotting

Wing discs (20 discs per lane) were lysed in 2× Laemmli Sample Buffer (Bio-Rad, #1610737) supplemented with protease inhibitor cocktail (Roche) and phosphatase inhibitor cocktail (Calbiochem). Protein samples were loaded in 4–15% gradient gels (Bio-Rad). Antibodies used for immunoblotting include mouse anti-GFP (1:1000; Cell Signaling Technology, #2955, RRID:AB_1196614) and as a loading control rabbit anti-GAPDH (1:5000; Santa Cruz Biotechnology, sc-25778, RRID:AB_10167668). Blots were visualized with fluorescent-conjugated secondary antibodies (LI-COR Biosciences) and the Odyssey Imaging System (LI-COR Biosciences).

## Laser ablation

Live imaging and laser ablation experiments were performed as previously described (*Rauskolb et al., 2014*). *en-Gal4 UAS-RFP/CyO; UAS-dcr2/TM6B* flies were crossed with *UAS-kstRNAi; Ubi-E-cad:GFP/ TM6B, UAS-a-specRNAi; Ubi-E-cad:GFP/TM6B* or *UAS-kstCRISPRa; Ubi-E-cad:GFP/TM6B* flies. Eggs were collected at 25°C for 4 hr and then shifted to 29°C for 88 hr. Wing disc culture was based on the procedure of *Dye et al., 2017*. A stock medium was prepared using Grace's medium (Sigma, G9771) without sodium bicarbonate but with the addition of 5 mM Bis-Tris and the pH was adjusted to 6.6–6.7 at room temperature. This was stored at 4°C for less than a month. Before every experiment, we added 5% fetal bovine serum (Thermo Fisher, 10082147), penicillin-streptomycin (Thermo Fisher, #15070063, 100× stock solution), and 10 nM 20-hydroxy-ecdysone (Sigma, H5142) to the medium. Larvae were floated on 25% sucrose and transferred into glass dishes with culture medium. Larvae of the desired genotype were selected and sterilized in 70% ethanol for 1 min. Then, we drew a circle on the glass bottom of a 35 mm glass-bottomed Petri dish (MatTek, P35G-0-14C) using glue made by mixing heptane with tape (Tesa, 5388). Wing discs were dissected out of larvae, transferred into this Petri dish, and oriented using tungsten needles. Then, we covered the discs with a Cyclopore Polycarbonate membrane (GE Health, 7060-2513) and glued it to the glass bottom to immobilize

discs. Discs were imaged every 0.2 s on a Perkin Elmer Ultraview spinning disc confocal microscope, and ablation of junctions was achieved using a Micropoint pulsed laser (Andor Technology) tuned to 365 nm. Paired cutting of junctions, one in the anterior compartment and another in the posterior compartment at a similar location, were performed and compared. The displacement of vertices for the first second after ablation was used to calculate the velocities.

## Quantification and statistical analysis

To obtain the surface of the wing disc and remove signals from the peripodial epithelium, we used the MATLAB toolbox ImSAnE (*Heemskerk and Streichan, 2015*) to detect and isolate a slice of the disc epithelium that surrounds the AJs, using E-cad as a reference, as described previously (*Pan et al., 2016*). The KstGFP, KstYFP, SqhGFP, and JubGFP images were created using ImSAnE.

For the fluorescence intensity heat maps, a custom MATLAB script was used (*Alégot et al., 2017*; *Alégot et al., 2019*; *Pan et al., 2018*). The script generates a 3D mask with the normalization channel (E-cad) keeping only the relevant pixels. The wing disc center (intersection between AP/DV boundaries) is picked for each image manually. Then, the picture is split into blocks of a given xy size ($3 \times 3$ $\mu m^2$), starting from the center, and the average intensity per pixel of each channel is measured. The intensity of the reference channel and the channel of interest are normalized over their respective average intensity. The ratio of the channel of interest over the reference channel is then determined. To average several discs, only matrices of the same xy size blocks were used. The center of the disc serves as a reference point; smaller matrices were expanded to correspond to the size of the biggest matrix and filled with NaN (Not-a-Number). We determined the minimum number of values required (usually three) to average the ratio for a given position. This means that the edges of the average disk are composed of the same minimum number of values, which corresponds to the n given for each experiment. Finally, signals from several wing discs were averaged and represented by the heat map, and a posterior versus anterior ratio was calculated.

Pearson's correlation coefficient was calculated to establish co-localization between different proteins by using the Coloc 2 Plugin for ImageJ (https://imagej.net/plugins/coloc-2).

Statistical significance was determined with GraphPad Prism software by performing Student's t-test (for comparison between two observations) or analysis of variance (ANOVA) with $p < 0.05$ set as the criteria for significance. The Dunnett test was used to derive adjusted p-values for comparisons against the control experimental value, and the Tukey test was used to derive adjusted p-values for multiple comparisons.

## Protein production and purification

To purify the actin binding domain of $\beta_H$-Spec fused with GST, we transformed BL21-DE3 cells (NEB) with pGEX-3X-ABDkst. Protein expression was induced with 0.2 mM isopropyl-β-D-thiogalactoside at room temperature for 12–15 hr. Cells were harvested and lysed by sonication in lysis buffer: PBS (pH 7.4), 1% Triton X-100, 5 mM dithiothreitol, 1 mM phenylmethyl sulfonyl fluoride, and complete Mini Protease Inhibitor Cocktail (Roche). After centrifugation, the supernatant was collected and passed through a 1 mL GST-Trap column (Cytiva) at 4°C. Then, the column was washed with PBS (pH 7.4) until the absorbance reached a steady baseline. To remove the GST-tag, the column was loaded with 80 units of Factor Xa (New England Biolabs, P8010) in cleavage buffer: 50 mM Tris-HCl, 150 mM NaCl, 1 mM $CaCl_2$, pH 7.5, and incubated for 16 hr at room temperature. To elute the ABD of $\beta_H$-Spec while removing the protease simultaneously, we used a HiTrap Benzamidine FF (Cytiva) column in tandem with the GST-trap column and eluted the ABD of $\beta_H$-Spec with cleavage buffer. To switch buffers, we dialyzed the protein against 25 mM HEPES pH 7.4, 150 mM NaCl, 1 mM TCP overnight. Then, the purified protein was concentrated by ultrafiltration to 30 $\mu M$ and stored at –80°C until used in experiments.

G-actin was prepared from rabbit muscle acetone powder as reported (*Lehrer and Kerwar, 1972*) and further purified with size exclusion chromatography on a Superdex 75 pg column (Cytiva, # 28989333) in buffer containing 5 mM Tris/HCl pH = 8, 0.2 mM $CaCl_2$, 0.5 mM ATP, 1 mM DTT. G-actin was polymerized to F-actin by the addition of 10× polymerization buffer (100 mM HEPES pH 7.0, 500 mM KCl, 20 mM $MgCl_2$, 10 mM EDTA). *Drosophila* nonmuscle myosin-2 subfragment-1-like protein (*Zip*, amino acids 1–813) was recombinantly overproduced together with the myosin

regulatory (*Sqh*) and essential light chain (*Mlc-c*) in the baculovirus/*Sf*9 insect cell system (Thermo Fisher Scientific) and prepared as described (*Heissler et al., 2015*).

## Co-sedimentation assays

For co-sedimentation assays, $\beta_H$-Spec CH domains or myosin were incubated with F-actin for 15 min at room temperature in an assay buffer containing 10 mM HEPES pH 7.4, 100 mM NaCl, 0.1 mM EGTA, 20 µM ATP, and 1 mM DTT and subsequently sedimented (100,000 × *g*, 15 min, 4°C, TLA-100 rotor) in an Ultima MAX-XP ultracentrifuge (Beckman). For competition assays, $\beta_H$-Spec and actin were preincubated for 15 min at room temperature before the addition of myosin. Supernatant and pellet fractions were separated and the pellet fraction was resuspended in an equal volume of assay buffer. Samples were supplemented with NuPAGE LDS sample buffer (Invitrogen, #NP0007) and heated for 10 min at 90°C. Supernatant and pellet fractions were resolved on 4–12% NuPAGE Bis-Tris polyacrylamide gels (Invitrogen, NP0323BOX). Gels were incubated with PageBlue protein staining solution (Thermo Scientific, #24620) and destained with water. Gels were documented with a ChemiDoc MP (Bio-Rad) and densitometric analysis was performed with Fiji (*Schindelin et al., 2012*). Data plots and secondary analysis were performed in Origin 2019.

## Model building

A homology model of the *Drosophila* myosin motor domain (*Zip*, amino acids 1–813) in the actin-bound state was modeled using the cryo-EM structure of the human nonmuscle myosin-2C motor domain in the rigor state (PDB entry: 5JLH) as a template. Both motor domains share ~72% sequence identity at the amino acid level. The motor domain model was built using Modeler (*Sali and Blundell, 1993*). The model of the *Drosophila* $\beta_H$-Spec calponin homology domain tandem (CH1-CH2, amino acids 1–278) was modeled using ColabFold (*Mirdita et al., 2022*). This model was superimposed onto the cryo-EM structure of the human β-III spectrin CH1 domain bound to F-actin (PDB entry: 6ANU) for binding site analysis.

# Acknowledgements

We thank Christian Klämbt for the β-Spec antibody, and Richard Ebright and Bryce Nickels for sharing equipment for protein purification. This research was supported by National Institutes of Health grants GM131748 (KDI) and R01GM143539 (KC).

# Additional information

## Funding

| Funder | Grant reference number | Author |
| --- | --- | --- |
| National Institute of General Medical Sciences | 131748 | Kenneth D Irvine |
| National Institute of General Medical Sciences | 143539 | Krishna Chinthalapudi |

The funders had no role in study design, data collection and interpretation, or the decision to submit the work for publication.

## Author contributions

Consuelo Ibar, Sarah M Heissler, Conceptualization, Formal analysis, Investigation, Methodology, Writing - original draft, Writing - review and editing; Krishna Chinthalapudi, Formal analysis, Investigation, Methodology, Writing - review and editing; Kenneth D Irvine, Conceptualization, Supervision, Funding acquisition, Methodology, Writing - original draft, Writing - review and editing

## Author ORCIDs

Consuelo Ibar http://orcid.org/0000-0002-1965-2696
Krishna Chinthalapudi http://orcid.org/0000-0003-3669-561X
Kenneth D Irvine http://orcid.org/0000-0002-0515-3562

Reviewer #1 (Public Review): https://doi.org/10.7554/eLife.84918.3.sa1
Reviewer #2 (Public Review): https://doi.org/10.7554/eLife.84918.3.sa2
Author Response: https://doi.org/10.7554/eLife.84918.3.sa3

## Additional files

### Supplementary files
• MDAR checklist

### Data availability
All data generated or analyzed during this study are included in the manuscript and supporting file.

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
