## [Editor Report · eLife assessment]

The manuscript provides **valuable** insights into the regulatory role and mechanisms of the spectrin cytoskeleton in mechanotransduction in *Drosophila*. The data are **compelling** in establishing that alpha and beta spectrin regulate the Hippo signaling pathway independently via their effect on cytoskeletal tension. The work will be of interest to cell and developmental biologists, particularly those who focus on mechanotransduction and the cytoskeleton.

---

## [Referee Report · Reviewer #1 (Public Review)]

Ibar and colleagues investigate the function of spectrin in *Drosophila* wing imaginal discs and its effect on the Hippo pathway and myosin activity. The authors find that both βH-Spec and its canonical binding partner α-Spec reduce junctional localization of the protein Jub and thereby restrict Jub's inhibitory effect on Hippo signaling resulting in activation of the Hippo effector Yorkie regulating tissue shape and organ size. From genetic epistasis analysis and analysis of protein localization, the authors conclude that βH-Spec and α-Spec act independently in this regulation. The major point of this study is that the apical localization of βH-Spec and myosin is mutually exclusive and that the proteins antagonize each other's activity in wing discs. *In vitro* co-sedimentation assays and in silico structural modeling suggest that this antagonization is due to a competition of βH-Spec and myosin for F-actin binding.

The study's strengths are the genetic perturbation that is the basis for the epistasis analysis which includes specific knockdowns of the genes of interest as well as an elegant CRISPR-based overexpression system with great tissue specificity. The choice of the model for such an in-depth analysis of pathway dependencies in a well-characterized tissue makes it possible to identify and characterize quantitative differences between closely entangled and mutually dependent components. The method of quantifying protein localization and abundance is common for multiple figures which makes it easy to assess differences across experiments. The flow of experiments is logical and in general, the author's conclusions are supported by the presented data. The findings are very well embedded into the context of relevant literature and both confronting and confirming literature are discussed.

The study shows how components of the cytoskeleton are directly involved in the regulation of the mechanosensitive Hippo pathway in vivo and thus ultimately regulate organ size supporting previous data in other contexts. The molecular mechanism regulating myosin activity by out-competing it for F-actin binding has been observed for small actin-binding proteins such as cofilin but is a new mode for such a big, membrane-associated actin-binding protein. This may inspire future experiments in different morphogenetic contexts for the investigation of similar mechanisms. For example, the antagonistic activity of βH-Spec and myosin in this tissue context might help explain phenomena in other systems such as spectrin-dependent ratcheting of apical constriction during mesoderm invagination (as the authors discuss). Against the classical view, the work shows that βH-Spec can act independently of α-Spec. Together the results will be of interest to the cell biology community with a focus on the cytoskeleton and mechanotransduction.

---

## [Referee Report · Reviewer #2 (Public Review)]

Ibar and colleagues address the role of the spectrin cytoskeleton in the regulation of tissue growth and Hippo signaling in an attempt to elucidate the underlying molecular mechanism(s) and reconcile existing data. Previous reports in the field have suggested three distinct mechanisms by which the Spectrin cytoskeleton regulates Hippo signaling and this is, at least in part, due to the fact that different groups have mainly focused on different spectrins (alpha, beta, or beta-heavy) in previous reports.

The authors start their investigation by trying to reconcile their previous data on the role of Ajuba in the regulation of Hippo signaling via mechanotransduction and previous observations suggesting that Spectrins affect Hippo signaling independently of any effect on myosin levels or Ajuba localization. Contrary to previous reports, the authors reveal that, indeed, depletion of alpha- and beta-heavy-spectrin leads to an increase in myosin levels at the apical membrane. Moreover, the authors also reveal that the depletion of spectrins leads to an increase in Ajuba levels.

---

## [Author Response]

The following is the authors' response to the original reviews.

We thank both reviewers for their comments, which have suggested changes that have improved the manuscript.

**Reviewer #1 (Public Review):**
[…] A weakness in the methodology is the link to tissue tension and conclusions about tissue mechanics. Methods that directly affect tissue tension and a more thorough and systematic application of laser ablation experiments would be needed to profoundly investigate mechanosensation and consequential effects on tissue tension by the various genetic perturbations.

Response: In revision, we have added some additional experiments that examine altered tension.

While the in-silico analysis of competing for F-actin binding sites for βH-Spec and myosin appears logical and supports the authors' claims, no point mutation or truncations were used to test these results in vivo.In its current structure the manuscript's strength, the genetic perturbations, is compromised by missing clear assessments of knockdown efficiencies early in the manuscript and other controls such as the actual effect on myosin by ROCK overactivation.

Response: In revision, we reorganized the manuscript and figures to document the knockdown efficiency earlier in the manuscript, and have added additional figure panels illustrating the effects of altered tension on myosin levels.

**Reviewer #2 (Public Review):**
[…] The authors suggest that Ajuba is required for the effect of beta-heavy spectrin. However, it is still formally possible that this could be a parallel pathway that is being masked by the strong phenotype of Ajuba RNAi flies.

Response: While it is formally true that the genetic requirement for Jub could reflect a role in parallel to, rather than downstream of, spectrins, our conclusion that spectrins act through Jub is based not only on the genetic requirement for Jub, but also on the influence of spectrins on junctional tension and Jub localization, which indicate that spectrins influence Jub activity in a manner consistent with their affecting the Hippo pathway through Jub.

One of the major points of the manuscript is the observation that alpha- and beta-heavy-spectrin are potentially working independently and not as part of a spectrin tetramer. This is mostly dependent on the observation that alpha- and beta-heavy-spectrin appear to have non-overlapping localizations at the membrane and the fact that alpha- and beta-heavy-spectrin localize at the membrane seemingly independently. It is not entirely obvious that a potential lack of colocalization and the fact that protein localization at the membrane is not affected when the other partner is absent is sufficient to argue that alpha- and beta-heavy-spectrin do not form a complex. Moreover, it is possible that the spectrin complexes are only formed in specific conditions (e.g. by modulating tissue tension).

Response: Our results argue that alpha- and beta-heavy-spectrin do not form a detectable complex in the wing disc under the conditions examined, and thus that they act independently is this context. However, we agree that it is possible that they could function together contexts, eg in other tissues or under different conditions, and we have revised the text in the Discussion to note this.

If indeed spectrins function independently, would it not be expected to see additive effects when both spectrins are depleted?

Response: Not necessarily, since both alpha- and beta-heavy-spectrin act through Jub, and there may be a limit as to how much Yki activity can be increased by Jub eg the increases in wing size induced by spectrin RNAi are similar to the increases in wing size observed with constitutive recruitment of Jub through alpha-catenin mutation (Alegot et al 2019).

Related to the two previous points, the fact that the authors suggest that both alpha- and beta-heavy-spectrin regulate Hippo signaling via Ajuba would be consistent with the necessity of an alpha- and beta-heavy-spectrin complex being formed. How would the authors explain that both spectrins require Ajuba function but work independently?

Response: The different spectrins both affect Jub because they both affect cytoskeletal tension, but our results suggest that they act in different ways to affect tension. We have made some revisions to the Discussion section to try to make this clearer.

Another major point of the manuscript is the potential competition between beta-heavy-spectrin and myosin for F-actin binding. The authors suggest that there is a mutual antagonism between the two proteins regarding apical F-actin. However, this has not been formally assessed. Moreover, despite the arguments put forward in the discussion, it seems hard to justify a competition for F-actin when beta-heavy-spectrin seems to be unable to compete with myosin. Myosin can displace beta-heavy-spectrin from F-actin but the reciprocal effect seems unlikely given the *in vitro* data.

Response: We show *in vivo*, *in vitro*, and in silico data that are all consistent with the inference that beta-heavy-spectrin and myosin compete for binding to F-actin. As the reviewer notes, and as we discuss, the *in vitro* competition experiments were limited because, for technical reason, we were unable to increase the protein concentrations higher. We also note that our *in vitro* experiments used an active form of myosin, which binds F-actin much more strongly than inactive myosin.

**Reviewer #1 (Recommendations For The Authors):**
While the flow of experiments is logical in general, I see major problems regarding the structure of the manuscript and essential controls:• It is very confusing to have samples (kst-CRISPRa) in figures 1-3 that were not introduced in the text until the second-last paragraph of the results. I would suggest introducing this elegant overexpression experiment early in the manuscript as it fits well in the scope of these experiments or alternatively (if the authors prefer) make a new figure containing all the data regarding the overexpression in the end.

Response: We have now moved these results to a new figure (new Fig 7) that is described later in the text.

• At the beginning of the manuscript, essential controls regarding the knockdown efficiency are missing in the main figure. Many of the key experiments are based on KD and as a reader, I want to assess their efficiency. Only in Figure 4, at the end of the manuscript, KST and α-Spec KD efficiency is revealed - this should be shown earlier and quantified properly. While reading the manuscript in its current form, the doubt remains that differences e.g. in α-Spec and KST KD can be explained by varying knockdown efficiencies as their levels can't be assessed.

Response: We have now moved these results to a new supplemental figure (Fig 1-supplement 1) that is cited earlier in the text.

• On a similar line, in Figure 5 where myosin activity is perturbed, induction or repression of myosin activity is only suggested but not formally shown. The authors have to demonstrate that this is indeed the case by showing the myosin signal, ideally accompanied by measurement of tissue tension.

Response: This was not included because we and others have assessed these manipulations in earlier publications. However, as requested we have now added a supplemental figure (Fig 6 supplement 1) showing myosin levels in these genotypes.

• On p. 7, the authors claim that "The epistasis of jub to kst suggests that βH-Spec regulates wing size through its tension-dependent regulation of Jub." While the authors show that KST KD increases myosin and junctional Jub, and that the wing overgrowth phenotype of KST KD depends on Jub, the tension-dependency was not demonstrated. To make that claim, the tension profile should be perturbed e.g. by overexpression of rok, myosin mutants (as the authors do in Fig 5) and the effect on Jub should be analyzed. Induction of tension in these conditions should be measured by laser ablation or a suitable alternative method. It might well be that the induction of Jub in KST KD is not via tension but an alternative mechanism such as the release of steric hindrance, interaction competition, etc. Also: Does KD of Jub affect spectrin localization?

Response: The effect of tension on Jub, and the effects of the myosin activity changes we employed on tension, have been analyzed in prior publications (eg Rauskolb et al 2014). To further address the issue raised by the reviewer here as to whether Kst affects Jub and wing growth via tension, we have also now added an additional experiment (Fig 3 supplement 1) in which we decreased tension in a βH-Spec RNAi wing disc by simultaneously expressing RNAi targeting Rok. The results show that the wing growth and Jub accumulation associated with βH-Spec RNAi are suppressed by Rok RNAi, consistent with our conclusion that these effects are mediated via cytoskeletal tension.

As KD of Jub alters the pattern of myosin accumulation in wing discs (Rauskolb et al 2019) it could be expected to have a complementary influence on βH-Spec localization, but we have not examined this.

• The authors make a very strong point in saying "The influence of βH-Spec on junctional tension is thus a direct consequence of its competition with myosin for overlapping binding sites on F-actin." While the authors provide some *in vitro* and in silico evidence, it was for example not possible to outcompete myosin by increasing levels of KST CH1-CH2 domains *in vitro* (for possible reasons the authors discuss). More importantly, the hypothesis that competition for actin binding is the definite cause of the antagonizing effect was not tested in vivo. Overexpression of a mutant version of KST that is unable to bind F-actin, or that has an increased affinity (etc) for actin was not tested. Such an experiment would be very valuable to enrich this manuscript but at least, claims like that have to be less bold and need to be written in a more speculative language.

Response: We consider creating and analyzing mutant forms of Kst in vivo to be beyond the scope of this manuscript, but as suggested we have now modified the text highlighted by the Reviewer to be more cautious.

Further points:• Why does the thickness of the wing disc epithelium change due to KST and α Spec KD, the authors should introduce this experiment better and draw a proper conclusion. Is there any relocalization of myosin along the apical-basal axis? Can the authors speculate about the differences between KST and α Spec KD?

Response: The epithelium thickness changes with α-Spec KD, but does not change with Kst KD. We think the explanation is provided by work from the Pan lab (done mainly in pupal eyes), which reported decreased cortical tension and increased apical area when α-Spec is lost. The interpretation in essence is that with the loss of attachment of F-actin to membranes along the lateral sides of the cells, the sides of the cells are "softer" and the cells expand laterally and thus also (by conservation of volume) shorten apical-basally. This is somewhat speculative, and it's not a focus of our study, but we have added some text to try to explain this better. Myosin along apical-basal axis was not visibly altered, but it is harder to analyze as it is very weak compared to junctional myosin.

• Given the authors' observation of differences in the relative localization of KST and α Spec (Figure 4), proper quantification of KST, α Spec and myosin levels along the apical-basal cell axis would be important. This would also ease data interpretation.

Response: We have now added a higher resolution image and also a line scan of Kst, α-Spec and Myo in a new supplemental figure (Fig 6 supplement 1)

• KD of α Spec seems to induce myosin activity more, causes a bigger reduction of wing thickness, a stronger induction of Jub, and a similar effect on wing size. What lead the authors to focus on KST rather than α Spec regarding the detailed analysis of myosin competition?

Response: Our observations identify a competition between Kst and myosin, but we have no indication that α-Spec competes with myosin. (It's conceivable that β-Spec might also compete with myosin in some contexts, but wing discs would not be a good place to examine this because the localization profiles of β-Spec and Myosin are so different).

• A big criticism regarding the figures is the bad color choice which makes it difficult to decipher the fluorescent signals. Likewise, the labels are difficult to read with the present coloring. They should really be changed.

Response: We have now changed the single color images to gray scale (for multi-color images we retain RGB coloring).

A minor point:• To make the manuscript more accessible for researchers outside the *Drosophila* field, I'd suggest adding explanatory labels for *Drosophila*-specific terms such as hyperactive myosin for sqhEE, a scheme to show where UAS-dcr2 is active, explain the purpose of Rfp expression as a control for tissue specificity, etc.

Response: We have added some explanations to the text to try to make this clearer.

**Reviewer #2 (Recommendations For The Authors):**
Major points:In lines 99-101, the authors mention that Deng et al., 2015 report that the depletion of spectrins leads to an increase in pMLC, with no associated changes in the colocalization of myosin and F-actin. It is more accurate to mention that Deng et al. suggest that the levels of a GFP-tagged rescue construct of MLC (Sqh) are unchanged in alpha-spectrin mutants, although this was not formally quantified. Moreover, there was not a formal assessment of colocalization between MLC and F-actin, but rather a suggestion that F-actin levels are unaffected by the alpha-spectrin mutation. Finally, Deng et al. mostly analyzed alpha-spectrin so it remains possible that the new results shown by the authors are compatible with the initial observations from Deng and colleagues.

Response: As suggested, we revised the text to note that Deng et al., 2015 specifically examined Sqh:GFP. While we agree that our focus is more on Kst and Deng et al focused on α-Spec, we also examined α-Spec, and as described our results examining Myosin and Jub differ from what was reported by Deng et al 2015.

As mentioned above, it is still possible that spectrins and Ajuba are working in parallel and Ajuba is not necessarily downstream of spectrins. The strong phenotype of Ajuba RNAi flies in adult wings could mask the effect of spectrins. Are the results similar in other settings, such as in the absence of Dicer2? Also, can Ajuba RNAi phenotypes be modified by overexpression of spectrins? This would provide further evidence of a link to Ajuba function.

Response: While formally it is true that the genetic requirement for Jub could reflect a role in parallel to, rather than downstream of, spectrins, our conclusion that spectrins act through Jub is based not only on the genetic requirement for Jub, but also on the influence of spectrins on junctional tension and Jub localization, which indicate that spectrins influence Jub activity in a manner consistent with their affecting the Hippo pathway through Jub.

We would not expect over-expression of spectrins in a jub RNAi background to further reduce Hippo signaling, and as the jub RNAi phenotype is much stronger than the Kst over-expression phenotype even if there were an effect it would likely be difficult to detect.

Regarding the potential independent functions of spectrins, it would be interesting to determine if alpha- and beta-heavy-spectrin can still interact at the level of the AJ despite the fact that their distributions appear to be partly non-overlapping. Would it be possible to assess this using PLA? If an interaction is not detected via PLA, it would be more convincing that spectrins are functioning independently.

Response: We have now performed this experiment, and no significant signal was detected by PLA. As a control, we used identical antibodies (GFP and α-Spec) to conduct PLA on α-Spec and β-Spec, and we did detect signal by PLA. These results (included in a revised Figure 4) further support the conclusion that α-Spec and βH-Spec are not physically associated in wing discs.

Related to this point, if the spectrins work independently, it is reasonable to assume that they could display additive effects. Is this the case? If alpha- and beta-heavy-spectrin are simultaneously depleted are the phenotypes more severe than either depletion alone?

Response: We disagree here. Since both alpha- and beta-heavy-spectrin act through tension and Jub, and there is likely a limit as to how much Yki activity can be increased by this pathway. For example, the increases in wing size induced by spectrin RNAi are similar to the increases in wing size observed with constitutive recruitment of Jub through alpha-catenin mutation (Alegot et al 2019), which may thus represent the maximum increase that can be induced through this pathway (as there are multiple, independent factors that regulate Hippo signaling).

Authors should modulate membrane tension and assess if this affects the localization of alpha- and beta-heavy-spectrin and, specifically, their colocalization, as their interaction could be regulated.

Response: As reported, we do see effects of tension on βH-Spec localization. We would not expect significant effects of membrane tension on α-Spec localization, but we consider analysis of this outside the scope of this manuscript.

In lines 185-187, the authors mention that beta-spectrin depletion does not affect beta-heavy-spectrin localization. Interestingly, Figure 4E appears to show that the levels of Kst-YFP appear to be lower in the beta-spectrin-depleted tissue. The localization of beta-heavy-spectrin is not necessarily affected but the overall levels could be.

Response: Indeed the levels appear slightly lower, but elucidating the reason for this will require further experiments that are beyond the scope of this manuscript (we suspect it is because cytoskeletal tension increases in β-Spec-depleted tissue as it does in α-Spec depleted tissue, which based on our observations should decrease levels of Kst at near junctions). The key point of these experiments was to show that α-Spec localization does not require βH-Spec, but does require β-Spec, which supports our conclusion that in wing discs α-Spec forms a complex with β-Spec but not with βH-Spec.

In lines 200-203, the authors state that beta-heavy-spectrin and myosin colocalize extensively at the apical region. However, this colocalization is not as clear as stated. Do the authors have alternative data that suggests that the two proteins are indeed colocalizing? Would it be possible to perform PLA to detect a potential colocalization?

Response: Unfortunately we do not have antibodies against both proteins that work well enough for PLA. However, we quantified the co-localization by analysis of Pearson's correlation coefficient, as reported in the manuscript. We also added an additional higher magnification image, and a line scan, in a supplemental figure (Fig. 6 supplement 1).

Authors should try to assess and quantify colocalization with F-actin for both beta-heavy-spectrin and myosin in wild-type conditions and when the levels (and/or activity) for each of them are modulated.

Response: We have added quantification of the co-localization of βH-Spec with F-actin and of myosin with F-actin to the revised manuscript.

Minor points:In lines 122-124, the authors should clarify the relevance of the observation that alpha-spectrin knockdown affects the thickness of the wing disc epithelium.

Response: We have added some text to try to elaborate on this.

In the intro, it is perhaps necessary to mention that there are conflicting reports regarding the role of spectrins in the regulation of cell proliferation, at least in the follicular epithelium. For instance, Ng et al., 2016 argued that spectrins do not regulate cell proliferation in FECs.

Response: Rather than wading into a detailed discussion of issues that are peripheral to this study, we modified the text in the Introduction to avoid implying that spectrins control cell proliferation in the ovary.

In Figures 1, 2, 3, and 4 (and respective supplements), it is encouraged that, wherever appropriate, the authors mark the different compartments or the relevant boundary using dashed lines, to more clearly indicate the regions to compare.

Response: We have now done this.

In Figure 2, supplement 1 panels C and D should have an indication of the genotype for clarity.

Response: We have now added this.

In lines 362-367, the authors suggest that other actin-binding proteins are likely to influence the role of beta-heavy-spectrin. Have the authors tested the role of spectrin interactors such as Ankyrin and Adducin?

Response: No, we have not examined this.